# Learning Visual Prior via Generative Pre-Training

Jinheng Xie[1]    Kai Ye[2*]    Yudong Li[2*]    Yuexiang Li[3]    Kevin Qinghong Lin[1]
Yefeng Zheng[3]    Linlin Shen[2]    Mike Zheng Shou[1†]

[1] Show Lab, National University of Singapore    [2] Shenzhen University
[3] Jarvis Research Center, Tencent YouTu Lab

{sierkinhane,mike.zheng.shou}@gmail.com

https://sierkinhane.github.io/visor-gpt

## Abstract

Various stuff and things in visual data possess specific traits, which can be learned by deep neural networks and are implicitly represented as the visual prior, *e.g.,* object location and shape, in the model. Such prior potentially impacts many vision tasks. For example, in conditional image synthesis, spatial conditions failing to adhere to the prior can result in visually inaccurate synthetic results. This work aims to explicitly learn the visual prior and enable the customization of sampling. Inspired by advances in language modeling, we propose to learn **Vis**ual pri**or** via **G**enerative **P**re-**T**raining, dubbed VISORGPT. By discretizing visual locations, *e.g.,* bounding boxes, human pose, and instance masks, into sequences, VISORGPT can model visual prior through likelihood maximization. Besides, prompt engineering is investigated to unify various visual locations and enable customized sampling of sequential outputs from the learned prior. Experimental results demonstrate the effectiveness of VISORGPT in modeling visual prior and extrapolating to novel scenes, *potentially motivating that discrete visual locations can be integrated into the learning paradigm of current language models to further perceive visual world.*

## 1  Introduction

The digital camera can continuously capture photographs of the visual world, such that tremendous photos and videos are currently shared on the Internet. In our world, various stuff and things possess specific *traits*, which have been correspondingly embedded in such visual data. In the current era of deep learning, deep neural networks [10, 40, 7] have demonstrated remarkable proficiency in learning from vast amounts of data, leading to the development of visual foundation models (VFMs) [15, 32, 43, 12, 22, 20]. Such *traits* have been accordingly learned and implicitly represented as the *visual prior* in VFMs, which has the potential to impact real-world applications. An example that highlights its importance can be seen in the field of image synthesis. To present high-quality and natural-looking images, the synthetic stuff and things must adhere to the visual prior such as the **spatial location, shape, and interaction of objects** (Fig. 1 (a)). A vivid example of layout-to-image is provided in Fig. 1 (b). When the spatial conditions do not adhere to the visual prior, such as the shape of 'donut' not being square, the size of 'person' being similar to that of 'donut', and 'donut' being floated in the air instead of being placed on 'dining table', the resulting synthetic contents may be inaccurate and visually inconsistent with the desired outcome. Despite recent advances in conditional image synthesis such as ControlNet [43] and GLIGEN [22], the challenge of continuously sampling customized spatial conditions that adhere to the visual prior remains a difficult problem, particularly for automatic synthesis of massive images with corresponding fine-grained annotations.

In this paper, we study the problem of how to explicitly learn visual prior from the real world and enable customization of sampling. If we would like to paint a series of instances on a canvas, we

---

* Equal Contribution    † Corresponding Author

37th Conference on Neural Information Processing Systems (NeurIPS 2023).

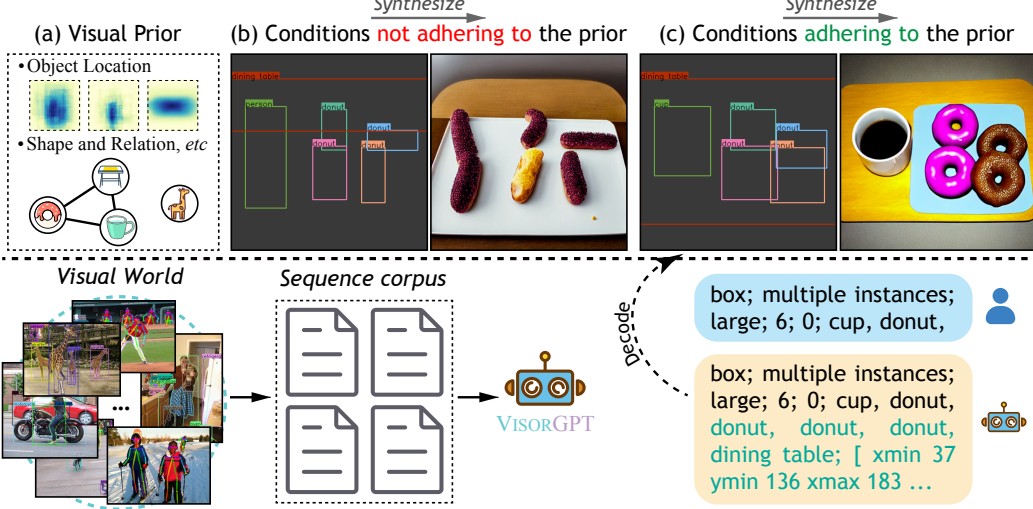

Figure 1: An overview of the problem of visual prior (top) and VISORGPT (bottom). (a) refers to visual prior, *e.g.,* location, shape, and relations of objects. (b) provides a *failure* case of image synthesis from spatial conditions that do not adhere to the prior. Specifically, the shape of the 'donut' not being square and 'donut' being floated in the air instead of being placed on 'dining table'. (c) displays a *success* case that conditions sampled from VISORGPT leads to a more accurate synthetic results. (d) illustrates that VISORGPT learns visual prior through sequence corpus converted from the visual world. (e) gives an example that a user customizes a sampling from VISORGPT by prompting.

should decide what to paint and also their shapes, locations, interactions, *etc.* It seems that these elements share a joint probabilistic prior, in which any stuff or things can be accordingly sampled to construct a scene. As there may be many potential variables in the prior, it is extremely hard to be comprehensively formulated. Over the past few years, significant advances have been made in language modeling [28, 29, 1, 6, 46], demonstrating their remarkable capacity for modeling the probabilistic distribution of sentences. Our focus is on learning the visual prior of location, shape, and relationships among categories, rather than raw pixels. It is possible to convert such visual information into a series of sequences, such that the visual prior can be learned by language modeling. To this end, as presented in Fig. 1 (d), we propose to learn **Vis**ual pri**or** via **G**enerative **P**re-**T**raining, dubbed VISORGPT. Thanks to the development of deep learning, many high-quality annotated data such as bounding-box [23, 38, 17], human pose [23, 21], instance mask [23] are publicly available. This provides sufficient location, shape, and relation information of stuff and things in the visual world. Since they are all encoded using 2D or 3D coordinates, we can simply convert them into a corpus of sequences. In this way, the visual prior can be learned by a pretext objective, *e.g.,* maximizing the likelihood of each sequence. Beyond this, prompt engineering is investigated to unify various visual locations and enable the customized sampling of sequential outputs from the learned prior.

As shown in Fig. 1 (e), according to the user's prompt, VISORGPT can correspondingly sample a sequence from the learned prior, which can be spatially decoded for image synthesis (Fig. 1 (c)). Since the decoded conditions adhere to the prior, the synthetic 'cup', 'dining table', and 'donut' are realistic and consistent with the desired semantics. This finding confirms that we can continuously customize spatial conditions from many aspects, *e.g.,* **data type, object size, number of instances, and classes**, using VISORGPT. With the advance of conditional image synthesis, it is feasible to generate an endless supply of synthetic images with their corresponding fine-grained annotations, potentially providing ample resources to train more robust and generalized visual intelligence models.

## 2 Related Works

**Language Modeling**. Language modeling aims to estimate the probability of a given sequence of words occurring in a sentence. In recent years, Large language models (LLMs) such as GPT series [28, 29, 1] and BERT family [6, 18, 26] have revolutionized the field of natural language processing. In particular, BERT family adopts the encoder(-decoder) architecture and employs masked language modeling techniques to model each given sentence bi-directionally in context. In contrast, GPT series employs the decoder-only architecture to sequentially model the probability of

next tokens by maximizing the likelihood of given sentences. Such a pretext objective allows for easy scaling up in terms of the model's parameters and training corpus. In this work, inspired by GPT series, we investigate the potential of a decoder-only architecture in modeling visual priors.

**Layout Generation**. Several related works [45, 4, 16, 2, 13] have focused on scene graphs or layout generation, specifically in the context of mobile interfaces and documents. For instance, Inoue *et al.* [13] addressed structural data using discrete diffusion models and provided a unified approach to diverse layout generation tasks. More recently, Feng *et al.* [8] directly delved into layout generation using LLMs via in-context learning. In contrast to these methods, our approach goes beyond the layout of interfaces and documents. We extend our focus to encompass various modalities, including human keypoints and semantic masks, and unify the modeling of this diverse information within a single straightforward learning objective.

**Conditional Image Synthesis**. With large-scale image-text datasets [36, 35], generative models [31, 30, 34, 32, 25, 24], *e.g.,* DALL-E, Imagen, and Stable Diffusion, have shown a significant capacity for synthesizing images of higher quality and greater diversity. Recently, more controllable image synthesis models such as training-based ControlNet [43] and GLIGEN [22] and training-free BoxDiff [42], have demonstrated a remarkable ability to control the synthetic contents precisely. When it comes to generating an extensive set of novel images, relying solely on spatial conditions from users or referring from images is inefficient. To tackle this problem, our VISORGPT is capable of continuously sampling customized and novel spatial conditions, making it possible to synthesize endless streams of data for various practical applications.

## 3 Methodology

In this section, we begin by presenting our problem formulation (§ 3.1) and prompt designs (§ 3.2) for unifying various visual information (*e.g.,* class and location) as textual sequences. Building upon this, we introduce our model architecture and pretext objective (§ 3.3) to model the visual prior. Finally, we provide practical examples of how to sample customized sequential outputs from VISORGPT (§ 3.4).

### 3.1 Problem Formulation

We assume that visual location $x$, *e.g.,* object bounding-box, human pose, and instance mask, follow a probabilistic prior distribution $p_x$. However, since $p_x$ is often unavailable in practice, this work aims to learn a model $f_\Theta$ with parameters $\Theta$ that can empirically approximate the latent probabilistic prior $p_x$. By doing so, we can sample new instances $\tilde{x}$ from the learned prior, denoted as $\tilde{x} \sim f_\Theta$, to facilitate various vision tasks, such as conditional image synthesis and action generation.

### 3.2 Visual Location as Sequences

In the *discrete* language domain, we witness that a variety of tasks, *e.g.,* translation and question-answering, can be integrated as a unified template (*e.g.,* prompts and responses) and then processed by one language model using different user prompts. However, when it comes to annotations of visual locations such as 2D object bounding-box, instance mask, and 3D human pose which are *continuous* and highly structured, a unified approach has yet to be explored and our objective is to investigate potential solutions to this issue. Following Chen *et al.* [3], we discretize visual annotations, *i.e.,* continuous numbers, into $m$ bins, such that location information can also be naturally represented as discrete tokens and $m$ integers are then accordingly added to the standard vocabulary. It means that coordinates can be represented by a sequence of words. In particular, each number representing visual localization will be quantified as an integer in the range of $[1, m]$. In this way, visual locations $x$ of each image can be then unified into a sequence $\mathbf{t} = \text{PROMPT}(x)$.

As visual annotations of various tasks are in different formats, we propose three universal prompts:

Prompt template $T_a$:
Annotation type; Data type; Size; #Instances; #Keypoints; Category names; Coordinates

Prompt template $T_b$:
Annotation type; Data type; Size; #Instances; #Keypoints; [Category name i Coordinate i]$_i$

Prompt template $T_c$:
Annotation type; Natural Language Input; Category names; Coordinates; Instruction; Category names; Coordinates

The provided prompts can be summarized in Tab. 1, which provides standardized templates to unify commonly used 2D and 3D visual location information into 1D textual sequences. Each prompt begins with the flags [Annotation type] and [Data type], which are the flags indicating the type of annotation and scene, *e.g.,* box and multiple instances. The follow-

Table 1: Candidate choices of prompt template.

| | |
|---|---|
| Annotation type | box; keypoint; mask; multimodal; · · · |
| Data type | object centric; multiple instances |
| Size | small; medium; large |
| #Instances | 1; 2; 3; · · · |
| #Keypoints | 14; 18 |
| Category name | cup; person; dog; · · · |

ing flags of [Size] and [#Instances] represent the average area and the number of instances in the current image, while [#Keypoints] indicates the number of keypoints annotated for each person, *i.e.,* 14 or 18. The following two flags are the [Category name] of each instance and their corresponding [Coordinate]. At last, [Natural Language Input] is extended for free-form language input and [Instruction] is used for iterative refinement of the current layout. Examples of these prompts are presented in the following sections. By employing our defined templates, we transform commonly used visual annotations into a large-scale sequential corpus. The corpus can be seamlessly ingested by language models, facilitating better learning of visual commonsense prior.

## 3.3 Learning Visual Prior via Generative Pre-Training

**Model Architecture**. In the past few years, many large language models have been successively proposed, such as GPT [28, 29, 1] and BERT [6, 18, 26] family, and recently introduced LLaMA [39]. We employ the GPT decoder-style transformer as our model to learn the visual probabilistic prior.

**Pretext Objective**. After processing the visual locations $x$ as textual sequences $t$ in § 3.2, we tokenize each sequence by byte-pair encoding (BPE) algorithm [37] to obtain a sequence with $n$ tokens $\mathbf{u} = \{u_1, u_2, \cdots, u_n\}$ such that a standard language modeling objective can be directly employed to learn visual prior by maximizing the following likelihood:

$$\mathcal{L} = \sum_i \log p(u_i | u_{i-k}, \cdots, u_{i-1}; \Theta), \tag{1}$$

where $k$ is the size of context window, and $p(\cdot|\cdot)$ indicates the conditional probability which is modeled by the neural network $\Theta$. Stochastic gradient descent is used to train the neural network.

## 3.4 Customizing Sequential Output

In addition to offering formatted visual annotations for learning a probabilistic prior, the standardized templates enable to *personalize sequential output for various applications through prompting*. For example, the customized sequential output can be employed as spatial conditions in image synthesis models (*e.g.,* ControlNet [43] and GLIGEN [22]). This opens up the possibility of synthesizing a broad range of data types to address diverse problems and challenges in computer vision. Here are a few representative scenarios:

**(a) Object Bounding-Box.** As we use a flag to distinguish different types of visual annotations, we can control the type of data and scene to be sampled from the learned probabilistic prior by setting the beginning tokens in the input prompt. Accordingly, we can set the beginning prompt as "box;" to generate sequential output with instances and corresponding bounding-box information. Besides, with flags like [Size], [#Instances], and [#Keypoints], we can sample a scene that adheres to multiple conditions. As depicted in Fig. 2 (a), we can input a prompt "box; multiple instances; small; 16; 0; kite, kite, person," as a prefix to require the VISORGPT to conditionally infer the remaining tokens. In this example, VISORGPT outputs the categories and their locations, specifically fulfilling the requirement of objects being in small size. In particular, (xmin, ymin) and (xmax, ymax) are special tokens indicating the top-left and bottom-right corners of the target object.

**(b) Human Pose.** With flags of [#Instances] and [#Keypoints], VISORGPT is capable of customizing sequential outputs involving instances with keypoints in a crowd scene. We give an example in Fig. 2 (b). Numbers (10 and 14) are added to the beginning of prompt as conditions to infer a scene consisting of 10 people with 14 keypoints. Note that, we use "a, b, c, d, · · ·" and "m0, m1, m2, m3, · · ·" as special tokens to distinguish each human keypoint and object boundary coordinate, respectively.

**(c) Instance Mask.** Beyond sparse coordinates as shown in (a) and (b), VISORGPT can deal with dense spatial annotations, *i.e.,* instance masks. Typically, pixel-level information can be represented

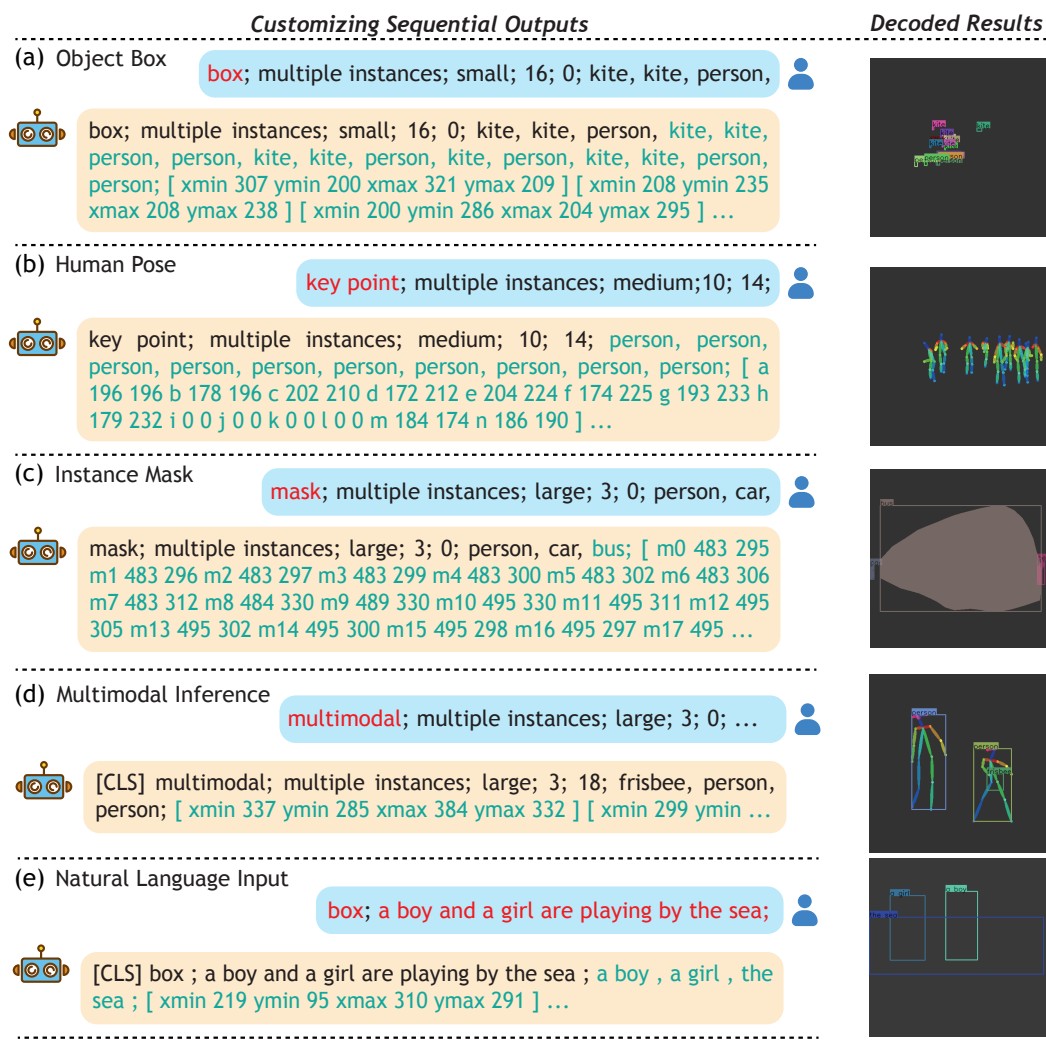

Figure 2: Examples of customizing sequential outputs from the proposed VISORGPT.

using a mask matrix or a set of boundary coordinates. For convenient sequentialization, we uniformly sample $n$ points along the angle in the polar space from object boundary coordinates to represent the pixel-level location, which is similar to [41]. We provide an example in Fig. 2 (c).

**(d) Multimodal Inference.** VISORGPT exhibits the capability of multimodal inference. In Fig. 2 (d), VISORGPT can deduce the presence of two individuals and a frisbee along with their bounding boxes and keypoints of these two people are also predicted at the same time.

**(e) Natural Language Input and Instruction.** Apart from the above prompts, VISORGPT can receive free-form language as input to generate corresponding layouts. We present an example in Fig. 2 (d). In addition, the proposed VISORGPT supports iterative refinement by instructions like "make the boy and girl closer".

## 4  Experiments

### 4.1  Experimental Setup

**Datasets.** We collect around 4 million sequences from the publicly available datasets for VISORGPT. In particular, we consider three types of commonly used visual annotations, *i.e.,* object bounding-box, human pose, and instance mask. In the MS-COCO dataset [23], we collect

Table 2: Training corpus for VISORGPT.

| Datasets (type) | #Categories | #Images |
| --- | --- | --- |
| Open Images (Box) [17] | 600 | 1,743,042 |
| Objects365 (Box) [38] | 365 | 1,728,775 |
| COCO (Box) [23] | 80 | 117,266 |
| ImageNet (Box) [33] | 1,000 | 38,285 |
| COCO (Keypoint) [23] | 1 | 53,473 |
| CrowdPose (Keypoint) [21] | 1 | 9,981 |
| COCO (Mask) [23] | 80 | 117,266 |
| Flickr30K [27] | 44,518 | 31,783 |
| Rico [5] | 25 | 35,851 |
| PubLayNet [44] | 5 | 315,757 |

~118K images annotated with 80 categories and their object bounding-boxes and instance masks. For each image, all object bounding-boxes and instance masks with their category information are formatted to a sequence, respectively. Beyond that, ~3.5 million bounding-box annotations of Objects365 [38] and Open Images [17] are also converted to sequences. Other types of annotations (*i.e.,* human keypoint) of MS-COCO (~54K) and CrowdPose (~10K) are also formatted to sequential data. For the object-centric scenario, we collect ~4K sequences from ImageNet-1K [33]. Besides, Flick30K [27] is employed to support free-form natural language as input. For the scenarios of mobile interface and document layout, we use the Rico [5] and PubLayNet [44] datasets. A summary is presented in Tab. 2.

**Evaluation Metrics**. We propose to evaluate VISORGPT from three aspects: **(i)** Evaluating the quality of sequences generated by VISORGPT. In the inference stage, as VISORGPT predicts sequences in the format given in § 3.2, it is necessary to examine whether the generated sequences can be decoded into visual locations. In particular, we generate a series of sequences using VISORGPT and calculate the accuracy whether it can be successfully decoded (termed **Format** in Table 5) and the number of categories matches the number of locations (termed **Matching** in Table 5). **(ii)** As discussed in § 3.2, we use flags, *i.e.,* [Size] and [#Instances], to indicate the average size and number of instances in the current sequence. Hence, we can control the average object size and the number of instances in the generated sequences via setting flags [Size] and [#Instances]. Then, we can calculate the accuracy whether the object size and the number of instances in the generated sequences are consistent with the given flags to validate the performance of controllability (termed **Size** and **#Instances**, respectively). **(iii)** Evaluating the learned probabilistic prior, *i.e.,* object location, shape, and relation among categories, on the *val* set of COCO, Objects365, and Open Images datasets. In this work, we propose to compare the discrete distribution of every visual prior. Specifically, to compute the **location** prior of a category, we initialize an empty canvas and convert the bounding-box of each instance of the category to a binary mask. Then, each mask is accumulated on the canvas and normalized as 2D location distribution. To compute the **shape** prior of a category, we calculate the ratio of width to height of each instance of the category, and estimate a discrete distribution as the shape prior. To establish the **relation** prior of a category to other categories, we count the number of co-occurrences between the category and other categories and estimate a discrete distribution. In this way, discrete prior of each category can be computed on COCO, Objects365, and OpenImages *val* sets as real one. Durring evaluation, we infer a series of sequences to compute the learned visual prior. Then we measure the similarity between learned and the real prior using the **Kullback-Leibler divergence** [14]. In addition, FID [11] is adopted to compare with layout generation methods [16, 13].

Table 3: Model card of VISORGPT.

| Models | #Parameters | #Training data | Annotation type | Batch size | Iterations | Learning rate | Sequence length $n$ |
|---|---|---|---|---|---|---|---|
| VISORGPT | 117M | 4M | box & keypoint & mask | 128 | 200K | $5.0e^{-5}$ | 1024 |
| VISORGPT† | 117M | 34K | box & keypoint & mask | 128 | 200K | $5.0e^{-5}$ | 1024 |

**Implementation Details**. We provide training details of VISORGPT in Tab. 3. VISORGPT adopted GPT-2 (base) architecture and was trained from scratch. We use all datasets reported in Tab. 2 to train VISORGPT , and Open Images and Objects365 are not involved to train VISORGPT†. In evaluation, each category is at least involved in ~80 valid predicted sequences by prompting (§ 3.2).

Table 4: Evaluation on training corpus scale and prompt templates of VISORGPT. The similarity between real probabilistic prior and the learned one is measured by KL divergence (KL Div).

| Models | Prompt | KL Div on COCO (↓) | | | KL Div on Open Images (↓) | | | KL Div on Objects365 (↓) | | |
|---|---|---|---|---|---|---|---|---|---|---|
| | | Location | Shape | Relation | Location | Shape | Relation | Location | Shape | Relation |
| VISORGPT† | $T_a$ | 1.133 | 1.483 | 0.452 | - | - | - | - | - | - |
| VISORGPT† | $T_a+T_b$ | 1.032 | 1.446 | 0.445 | - | - | - | - | - | - |
| VISORGPT | $T_a$ | 1.212 | 1.813 | 0.561 | 0.890 | 2.775 | 3.715 | 1.969 | 1.345 | 2.790 |
| VISORGPT | $T_a+T_b$ | 1.583 | 1.710 | 0.581 | 1.007 | 2.782 | 3.888 | 1.995 | 1.377 | 2.765 |

## 4.2 Quantitative Results

**Evaluation on Learned Visual Prior**. In Tab. 4, we present the measured similarity between real probabilistic prior and the one learned by VISORGPT on the validation sets of COCO, Open Images, and Objects365, using KL divergence. The prompt template $T_a$ and $T_a+T_b$ in § 3.2, are

Table 5: Evaluation on customized outputs (%).

| Datasets | Quality (↑) | | Controllability (↑) | |
|---|---|---|---|---|
| | Format | Matching | Size | #Instances |
| COCO | 100.0 | 100.0 | 92.02 | 100.0 |
| Open Images | 99.97 | 99.40 | 89.35 | 98.71 |
| Objects365 | 99.99 | 99.94 | 91.52 | 99.78 |

used for comparison. Overall, VISORGPT $T_a$ and $T_a+T_b$ exhibit comparable performance, indicating both prompt templates have comparable capability for learning visual prior.

**Evaluation on Customized Sequences**. We present the quality of generated sequences and the performance of VISORGPT's controllability in Tab. 5. It is obvious that nearly all predicted sequences can be decoded successfully in three datasets. Additionally, in over 99% of sequences, all instances can match their respective locations. Besides, the table shows that VISORGPT achieves accuracies of 92.02%, 89.35%, and 91.52% in controlling the average object size on COCO, Open Images, and Objects365 datasets, respectively. Furthermore, VISORGPT can achieve an accuracy of over 98% in controlling the number of instances across all three datasets. These findings demonstrate the strong capacity of VISORGPT in reasoning high-quality sequences and control the object size and number of instances in the scene.

**Comparison with Layout Generation Methods.** We also conduct experiments to compare with layout generation methods and the corresponding results are presented in Table 6. The experimental results include FID and Align. score (referred from LayoutDM [13]) with lower values indicates better performance. Notably, VISORGPT significantly surpasses these state-of-the-art methods with better FID and Align.

Table 6: Comparison to previous methods.

| Methods | Rico | | PubLayNet | |
|---|---|---|---|---|
| | FID ($\downarrow$) | Align. ($\downarrow$) | FID ($\downarrow$) | Align. ($\downarrow$) |
| MaskGIT [2] | 52.1 | 0.015 | 27.1 | 0.101 |
| BLT [16] | 88.2 | 1.030 | 116 | 0.153 |
| BART [19] | 11.9 | 0.090 | 16.6 | 0.116 |
| VQDiffusion [9] | 7.46 | 0.178 | 15.4 | 0.193 |
| LayoutDM [13] | 6.65 | 0.162 | 13.9 | 0.195 |
| VISORGPT (Ours) | **5.85** | **0.109** | **9.18** | **0.103** |

score on Rico and PubLayNet datasets, showcasing the superior capability of VISORGPT to model and generate layouts.

## 4.3 Visualization Results

['bottle, dining table, person, knife, bowl, bowl, oven, person, cup, cup, bowl, bowl, broccoli, spoon']   ['potted plant, chair, chair, dining table, refrigerator, banana, oven, sink, orange, orange, orange, orange, chair, orange']

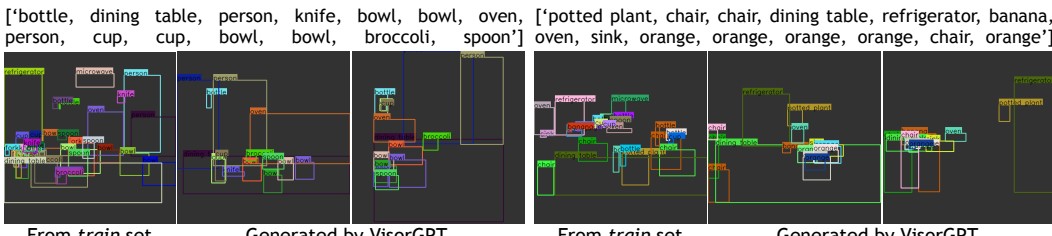

From *train* set     Generated by VisorGPT     From *train* set     Generated by VisorGPT

Figure 3: Generation diversity of VISORGPT.

**Generation Diversity**. One of the concerns is that VISORGPT would memorize the overall data distribution and cannot generate diverse and novel layouts. We present examples in Fig. 3 to address the concern. Firstly, we specify target classes (*e.g.*, "bottle, dining table, person, knife, bowl, oven, person, cup, cup, bowl, bowl, broccoli, spoon") and then select layouts satisfying these conditions from COCO *train* set (left in Fig. 3, only one satisfied sample over 110K samples). In contrast, we use the same target classes as prompts for VISORGPT to infer their bounding boxes. As depicted in Fig. 3, the bounding boxes generated by VISORGPT exhibit diversity and considerable differences from those selected ones from COCO *train* set.

**Scene Completion**. VISORGPT can receive user conditions and reasonably complete the corresponding missing classes in an interactive way, which can also demonstrate the capability of novel/unseen layout generation. We present examples in Fig. 4. One can observe that users can roughly draw two instances of "person" across the canvas (above) and prompt VISORGPT to deduce the remaining "skis" (below). To avoid cherry-picking, we provide the distribution of generated "skis" using 100 samples when given various scenes.

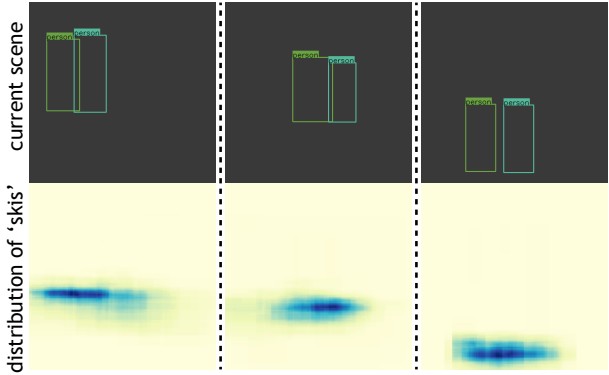

Figure 4: Scene completion.

As illustrated, when the two "person" vary from left-top to middle-bottom, the distribution of "skis"

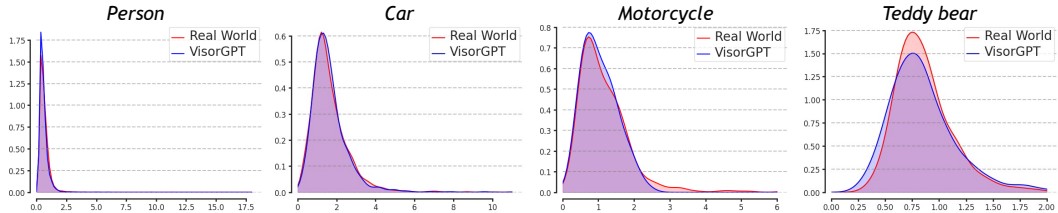

Figure 6: Shape prior of the categories of 'person', 'car', 'motorcycle', and 'teddy bear'.

exhibits consistent variations. This validates that VISORGPT does not rely on memorization. Instead, it learns intrinsic visual priors among categories, empowering it to infer novel/unseen layouts, accommodating the diverse requirements of users.

**Relation Prior.** Fig. 5 illustrates the comparison between the real-world relation matrix among 30 categories and the one estimated by VISORGPT. Each row depicts the relation prior of one category to others. For instance, it can be observed from the real world matrix that the 'person' (the first row) frequently interacts with other categories such as 'dog' and 'cat'. Similarly, in the

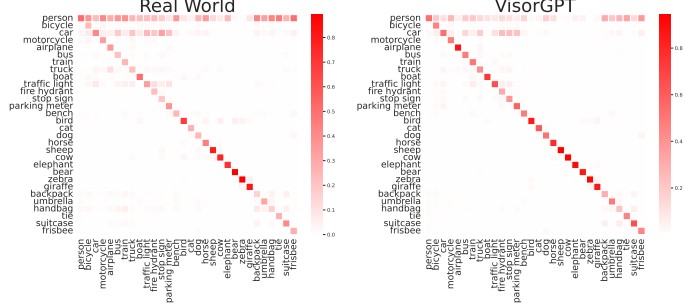

Figure 5: Relation matrix among 30 categories on COCO.

third row, the co-occurrence between 'car' and 'bus', 'truck', and 'stop sign' is larger than that of other categories. Notably, it is clear that the relation prior learned by VISORGPT is very close to that of the real-world one. This indicates that VISORGPT can capture the real relationships among categories and generate sequential output that aligns with these visual prior.

**Location Prior.** In addition to the quantitative results presented above, we visualize the comparison between the location prior learned by VISORGPT and the real one across various categories. Fig. 7 displays the location prior of three categories, including 'surfboard', 'tie', and 'train'. It is noticeable that, in each column, the location prior learned by VISORGPT is similar to the real one. For instance, from the first column, one can observe that the real distribution of 'tie' is mainly located in the lower-middle region, and the shape prior learned by VISORGPT exhibits a similar pattern.

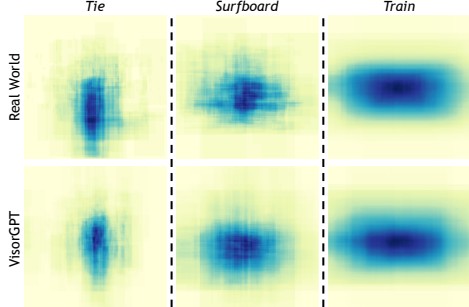

Table 7: Location prior of some categories.

**Shape Prior.** Fig. 6 shows the shape prior of four categories, such as 'person' and 'motorcycle'. To facilitate comparison, we employ kernel density estimation to estimate a continuous distribution from the discrete one. We observe that the shape prior learned by VISORGPT is close to those of the real visual world. For example, in the real world, the ratio of width to height of a car is almost always larger than 1, and the estimated shape prior of 'car' is mainly distributed around 1.8. It is evident that the learned probabilistic prior by VISORGPT, represented by the blue line, closely approximates the real one, represented by the red line. Overall, the shape priors of other categories learned by VISORGPT well match that of the real world.

## 4.4 Ablation Studies

Tab. 8 presents the impact of Special Words (SW), Textual Knowledge (TK, *i.e.,* with model weights initialized from the official pre-trained GPT-2), the number of sequences (#Seq), and model size (#Param). **(a)** Results on Tab. 8a are measured by the average KL divergence of location and shape prior. This confirms that the special words can potentially improve VISORGPT's performance in learning the visual prior. Notably, we found that the NLP textual knowledge deteriorated the performance of VISORGPT. We attribute this to the fact that the association between visual coordinates

Table 8: Impact of Special Words (SW), Textual Knowledge (TK), Number of Sequences (#Seq), and Model size (#Param). We use KL Div ($\downarrow$) as evaluation metric.

| (a) Effect of SW and TK. | | | | (b) Effect of #Seq. | | | (c) Effect of #Param. | |
| --- | --- | --- | --- | --- | --- | --- | --- | --- |
| SW | TK | COCO | Open Images | #Seq | COCO | Open Images | #Param | COCO |
| ✓ | ✗ | **1.647** | **1.895** | ~40 | 1.958 | 3.247 | 117M | 0.850 |
| ✗ | ✗ | 1.720 | 1.950 | ~80 | 1.195 | 2.460 | 345M | 0.836 |
| ✗ | ✓ | 1.959 | 2.240 | ~120 | **0.930** | **2.144** | 762M | **0.798** |

and natural language is relatively weak, thus it becomes inessential to learn visual prior from visual annotations. **(b)** In Tab. 8b, we find that increasing the number of sampled sequences leads to a more precise estimation of the visual prior by VISORGPT. **(c)** In Tab. 8c, we investigate the impact of model size on learning visual prior. For simplicity and efficiency, we replace VISORGPT architecture by three GPT versions and train it using only COCO (box) data. The results demonstrate the scalability of VISORGPT, *i.e.,* modeling the visual prior better with increased learnable parameters.

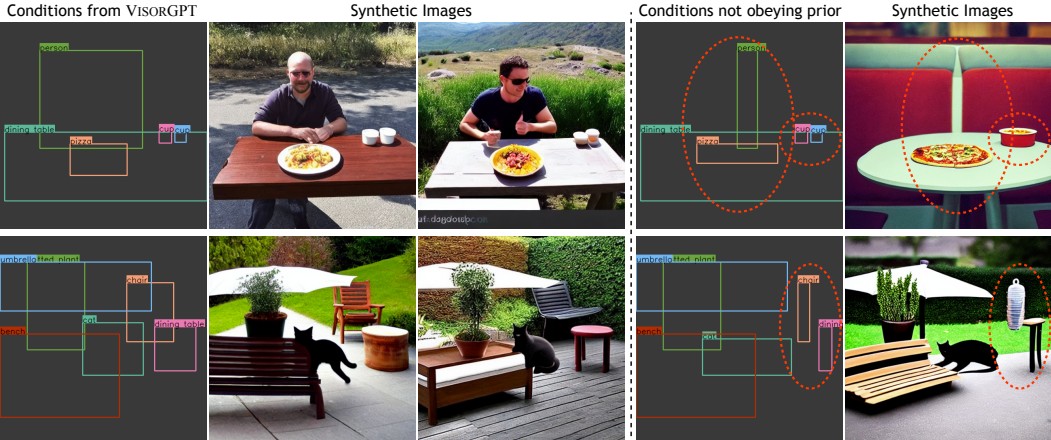

Figure 7: Comparison of synthetic images from *object boxes* adhering to the prior (left) or not (right).

## 4.5 Applications

**Conditional Image Synthesis**. VISORGPT's remarkable ability to infer visual categories and their locations based on user-customized prompts shows promising potential for generating customized images that still maintain a sense of realism. Here, we utilize ControlNet [43] and GLIGEN [22] to synthesize images from keypoints and bounding-boxes, respectively. We showcase some examples in Fig. 7 and 8. The first and fourth columns in Fig. 7 present the customized spatial conditions sampled from VISORGPT and the conditions not adhering to the visual prior. The second, third, and fifth columns provide synthetic results by GLIGEN conditioned on the corresponding spatial conditions. For example, on the first three columns, it is evident that the spatial conditions sampled from VISORGPT are more natural, such that the synthetic images are realistic and natural-looking. However, when the conditions (the last two columns) do not adhere to the prior, such as 'person' not being on a similar scale to 'dining table', the width of 'pizza' being too long, and the width of 'chair' being too short, the synthetic contents like 'person', 'chair', and 'dining table' appear abnormal, also impacting the authenticity of other objects like the two cups (circled in red dotted line).

Moreover, VISORGPT is capable of inferring sequences that include instances with keypoint information. For example, as shown in Fig. 8, we can provide a prompt like "key point; multiple instances; large; 13; 18; person, " to VISORGPT. This allows it to conditionally imagine a scene involving 13 people with their keypoint coordinates. Decoded results can be used as spatial conditions for image synthesis by ControlNet (shown in the last two columns). More examples can be found in supp.

## 5 Conclusion

This work proposed a novel approach, VISORGPT, to explicitly learning the probabilistic prior of the visual world through generative pre-training. This was achieved by transforming the continuous visual locations into discrete tokens by prompting and training a transformer decoder to maximize

**Input prompt:**
key point; multiple instances; large; 13; 18; person,

**Output sequence:**
key point ; multiple instances ; large ; 13 ; 18 ; person , person , person , person , person , person , person , person , person , person , person , person ; [ a 252 158 b 260 182 c 276 184 d 280 222 e 0 0 f 244 181 g 0 0 h 0 0 i 265 247 j 0 0 k 0 0 l 231 244 m 0 0 n 0 0 o 259 154 p 250 153 q 269 156 r 0 0 ] [ a 172 163 b 174 199 c 200 198 d 212 236 e 206 279 f 147 199 g 135 238 h 132 284 i 195 280 j 200 327 k 210 380 l 158 280 m 161 330 n 164 382 o 176 160 p 166 160 q 186 166 r 159 166 ] [ a 356 165 b 374 200 c 396 198 d 412 245 e 400 295 f 352 201 g 347 240 h 344 282 i 395 284 j 403 345 k 412 405 l 364 285 m 371 347 n 380 404 o 361 162 p 0 0 q 376 166 r 0 0 ] [ a 302 170 b 300 194 c 318 191 d 0 0 e 0 0 f 282 195 g 278 233 h 282 267 i 323 272 j 0 0 k 0 0 l 297 272 m 0 0 n 0 0 o 304 165 p 296 166 q 0 0 r 287 173 ] [ a 252 169 b 252 195 c 268 194 d 0 0 e 0 0 f 235 196 g 0 0 h 0 0 i 268 261 j 0 0 k 0 0 l 240 260 m 0 0 n 0 0 o 257 165 p 248 166 q 262 168 r 244 171 ] [ a 92 177 b 93 203 c 116 204 d 0 0 e 0 0 f 71 202 g 0 0 h 0 0 i 0 0 j 0 0 k 0 0 l 0 0 m 0 0 n 0 0 o 97 174 p 87 175 q 104 177 r 82 177 ] [ a 132 144 b 121 177 c 148 178 d 0 0 e 0 0 f 94 177 g 83 233 h 91 281 i 140 264 j 140 333 k 138 398 l 104 266 m 104 334 n 104 400 o 138 140 p 128 140 q 0 0 r 114 142 ] [ a 248 181 b 249 210 c 272 210 d 283 243 e 269 280 f 227 209 g 216 246 h 216 282 i 263 290 j 259 340 k 251 393 l 234 289 m 233 337 n 235 390 o 252 176 p 244 177 q 260 181 r 237 181 ] [ a 348 200 b 352 223 c 372 222 d 0 0 e 0 0 f 332 223 g 327 256 h 0 0 i 366 291 j 364 336 k 362 386 l 340 292 m 341 336 n 343 385 o 350 196 p 343 196 q 357 195 r 340 200 ] [ a 211 172 b 212 203 c 232 202 d 0 0 e 0 0 f 192 204 g 0 0 h 0 0 i 228 269 j 226 323 k 226 370 l 206 269 m 204 323 n 198 369 o 214 169 p 207 170 q 0 0 r 199 174 ] [ a 293 156 b 292 178 c 310 177 d 0 0 e 0 0 f 273 178 g 0 0 h 0 0 i 0 0 j 0 0 k 0 0 l 0 0 m 0 0 n 0 0 o 296 154 p 290 154 q 303 156 r 0 0 ] [ a 484 174 b 20 90 c 0 0 d 0 0 e 0 0 f 471 186 g 468 229 h 462 271 i 500 251 j 500 306 k 503 353 l 473 251 m 470 307 n 469 357 o 489 170 p 481 170 q 500 167 r 476 169 ] [ a 40 199 b 42 236 c 70 234 d 76 278 e 0 0 f 12 238 g 4 284 h 25 301 i 61 304 j 53 364 k 48 425 l 26 306 m 23 364 n 20 423 o 46 197 p 34 197 q 56 204 r 27 201 ]

**Input prompt:**
key point; multiple instances; large; 3; 18; person,

**Output sequence:**
key point; multiple instances; large; 3; 18; person, person, person; [ a 364 210 b 401 258 c 357 258 d 320 327 e 320 290 f 446 258 g 436 333 h 388 331 i 348 367 j 243 369 k 0 0 l 408 376 m 0 0 n 0 0 o 364 203 p 373 203 q 0 0 r 404 207 ] [ a 169 203 b 170 248 c 121 248 d 81 344 e 131 279 f 221 248 g 254 327 h 192 257 i 128 385 j 64 396 k 0 0 l 215 388 m 215 399 n 0 0 o 155 192 p 184 195 q 136 207 r 196 209 ] [ a 256 188 b 277 235 c 237 232 d 0 0 e 0 0 f 316 237 g 330 307 h 262 326 i 247 363 j 0 0 k 0 0 l 301 366 m 0 0 n 0 0 o 254 180 p 267 182 q 0 0 r 287 193 ]

**Input prompt:**
key point; multiple instances; large; 4; 18; person,

**Output sequence:**
key point; multiple instances; large; 4; 18; person, person, person, person; [ a 0 0 b 252 87 c 504 258 d 508 281 e 0 0 f 0 0 g 0 0 h 0 0 i 509 302 j 509 340 k 508 372 l 0 0 m 0 0 n 0 0 o 0 0 p 0 0 q 0 0 r 0 0 ] [ a 412 227 b 420 265 c 445 267 d 464 313 e 435 329 f 395 263 g 360 301 h 407 298 i 436 372 j 0 0 k 0 0 l 398 370 m 0 0 n 0 0 o 424 221 p 403 220 q 439 225 r 396 227 ] [ a 254 167 b 233 197 c 272 181 d 329 176 e 374 171 f 194 213 g 184 267 h 161 333 i 265 321 j 285 396 k 0 0 l 229 320 m 241 387 n 0 0 o 254 160 p 243 163 q 0 0 r 224 169 ] [ a 0 0 b 10 78 c 490 217 d 500 268 e 0 0 f 0 0 g 0 0 h 0 0 i 496 305 j 501 377 k 0 0 l 0 0 m 0 0 n 0 0 o 0 0 p 0 0 q 498 186 r 0 0 ]

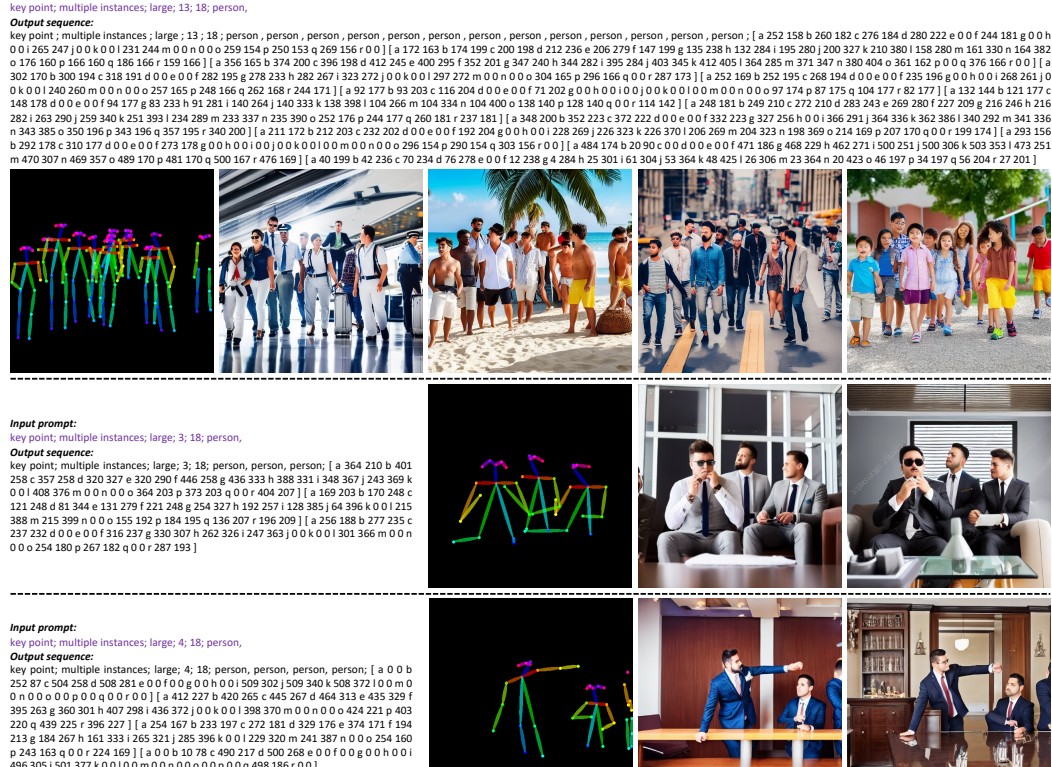

Figure 8: Illustration of input prompts (comprising multiple instances with *keypoints*), output sequences, decoded results and synthetic images.

the likelihood of training sequences. As a result, VISORGPT exhibits significant potential in comprehending real-world visual prior and leveraging this knowledge to create plausible scenes under a variety of customized prompts. This ability can facilitate the automatic synthesis of a vast number of images, along with their corresponding fine-grained annotations, using ControlNet and GLIGEN. This could potentially yield ample resources to train more robust visual intelligence models.

**Broader Impact**. This work demonstrates the substantial capacity of Large Language Models (LLMs) to effectively model spatial visual locations through language modeling. In the subsequent generations of LLMs, the visual priors discussed here can be seamlessly integrated into large-scale training, endowing LLMs with the ability to possess not only textual knowledge but also the capability to perceive and understand the visual world. While current datasets may have limited annotations for object categories, the vast knowledge present in extensive text corpora allows for the association of known objects with novel ones through language-based knowledge, such as synonyms. This presents a promising direction for the next generation of LLMs to expand their understanding and modeling of our visual world.

**Limitation**. We encountered limitations regarding the number of instances that could be included in each sequence due to the maximum token length, despite converting each mask annotation to a fixed length. In the future, we plan to incorporate large-scale natural language corpora into training and extend the maximum sequence length.

## Acknowledgment

This project is supported by the National Research Foundation, Singapore under its NRFF Award NRF-NRFF13-2021-0008, and the Ministry of Education, Singapore, under the Academic Research Fund Tier 1 (FY2022).

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

# 6 Appendix

## 6.1 Advantages compared to selecting GT bounding boxes from *train* set

**(i) Diverse layouts generation**. When users are required to specify a large number of classes (larger than 10), the layouts selected from COCO *train* set including all such classes are limited. As shown in Fig. 3, when target classes ('bottle', 'dining table', 'person', 'knife', 'bowl', 'bowl', 'oven', 'person', 'cup', 'cup', 'bowl', 'bowl', 'broccoli', 'spoon') are considered, there is only one relevant layout containing all the target classes from COCO *train* set. In contrast, as shown in Fig. 3, VISORGPT can receive these target classes to infer a lot of corresponding layouts with much diversity.

**(ii) User-customized / Controllability**. Compared to selecting from ground truth bounding boxes, one of the main advantages of VISORGPT lies in its ability to extrapolate novel layouts based on users' configurations, as depicted in Fig. 10 (c). VISORGPT can deduce the missing classes' positions ('skis, skis, skis, skis') based on the layout ('person, person') given by users.

**(iii) Free-form captions as layout condition**. Incorporating captions in the training sequences enables VISORGPT to receive free-form natural language input as a condition (*e.g.,* a boy and a girl are playing by the sea or a train is running toward the man) and generate corresponding layouts (as shown in Fig. 10 (b)).

## 6.2 Extrapolation ability

To showcase VISORGPT's extrapolation ability, we present examples in Fig. 10 (a). As demonstrated, when the 'sea' occupies the majority of the canvas, the 'surfboard' can appear throughout the entire canvas. Upon shifting the 'sea' region to the middle part of the canvas, the 'surfboard' is accordingly confined within the middle region. This phenomenon demonstrates that VISORGPT does not solely rely on memorization; rather, it learns intrinsic relation, location, and shape priors among categories. Consequently, VISORGPT can extrapolate unseen/novel layouts based on users' configurations.

## 6.3 Examples of Training Sequences

Here, we give some examples of various types of training sequences on different datasets:

Human Pose (COCO):
key point; multiple instances; large; 1; 18; person; [ a 190 120 b 266 146 c 318 143 d 385 232 e 338 269 f 214 150 g 0 0 h 0 0 i 312 280 j 365 296 k 359 420 l 258 283 m 194 344 n 301 383 o 197 100 p 181 103 q 234 84 r 0 0]

Human Pose (CrowdPose):
key point; multiple instances; large; 2; 14; person, person; [ a 312 201 b 306 200 c 311 232 d 269 214 e 298 257 f 231 206 g 296 275 h 307 275 i 251 244 j 271 235 k 274 292 l 283 295 m 304 153 n 310 191] [ a 179 247 b 165 245 c 164 313 d 160 315 e 221 316 f 207 279 g 155 343 h 144 366 i 242 337 j 240 367 k 210 431 l 300 418 m 172 176 n 177 227] key point; multiple instances; large; 2; 14; person, person; [ a 240 178 b 304 168 c 228 239 d 0 0 e 261 236 f 0 0 g 251 296 h 289 296 i 0 0 j 0 0 k 0 0 l 0 0 m 261 92 n 272 156] [ a 314 160 b 363 158 c 274 232 d 356 264 e 224 260 f 271 263 g 298 315 h 341 324 i 0 0 j 332 442 k 0 0 l 0 0 m 287 64 n 333 133]

Instance Mask:
mask; multiple instances; medium; 1; 0; clock; [ m0 224 291 m1 226 299 m2 227 306 m3 228 313 m4 233 320 m5 238 325 m6 245 329 m7 252 332 m8 259 334 m9 266 335 m10 274 333 m11 281 330 m12 288 327 m13 293 323 m14 299 318 m15 303 312 m16 305 305 m17 307 298 m18 310 291 m19 308 284 m20 307 276 m21 303 269 m22 299 263 m23 295 257 m24 288 254 m25 280 251 m26 273 250 m27 266 249 m28 259 249 m29 252 251 m30 246 256 m31 240 260 m32 235 265 m33 229 270 m34 227 277 m35 225 284]

Object Centric Bounding-Box:
box; object centric; large; 1; 0; castle; [ xmin 236 ymin 142 xmax 413 ymax 232]

## 6.4 Implementation Details

All experimental evaluations were conducted on eight NVIDIA Tesla V100-32GB GPUs using PyTorch. In order to include special words, we created a new vocabulary containing a total of 30,769 words based on a standard vocabulary. To optimize computational efficiency and memory utilization, we utilized the DeepSpeed framework. To serialize visual locations, we first resized the long side of each image to a length of 512 pixels

and then shifted the image content to the center by padding the short side to a length of 512 pixels. As a result, the number of bins $m$ was set to 512. The flag of [Size] indicates the average area of all instances in the image and we set the flag according to the rule:

$$\begin{cases} \text{"small"} & \text{average area} < 32^2 \\ \text{"medium"} & 32^2 \leq \text{average area} < 96^2 \\ \text{"large"} & \text{average area} \geq 96^2 \end{cases}.$$

We omitted person instances with fewer than five keypoints. To enable continuous generation, we designed and trained models based on the prompt format (b). Specifically, VISORGPT$^\dagger$ (a&b) and VISORGPT (a&b) were trained using the same number of sequences as VISORGPT$^\dagger$ (a) and VISORGPT (a), respectively. The only difference is that we randomly utilized prompt format (a) or (b) to construct each training sequence.

During the evaluation stage, we set the maximum sequence length of our model (VISORGPT) to 256 tokens to ensure efficient inference. In the ablation studies, we added special words only to the [Coordinate] term, and we reported the average KL divergence between the location and shape priors learned by VISORGPT and those in the real world. Since training large-scale language models is time- and resource-consuming, we trained only three types of VISORGPT with respect to GPT-2 (base, medium, large) with a maximum token length of 256 in 50,000 iterations on COCO (Box) data.

## 6.5 Evaluation Details

To estimate discrete visual prior from VISORGPT, we infer a series of sequences via prompting as below:

```
Code in Python:
f"box; multiple instances; random.choice(['small', 'medium', 'large']);
random.randint(2, 10); 0; category name,"
```

To ensure that each category in a given dataset is sufficiently represented in the sequence data used for estimating the visual prior, we specify a minimum number of sequences in which each category must appear. Table 9 provides an overview of the predicted sequences that are used for evaluation.

Table 9: Details about the predicted sequences for evaluation.

| Datasets | #Categories | #Predicted Seq. | Min #Seq. Per Category |
|----------|-------------|-----------------|------------------------|
| Open Images (Box) | 600 | 48,000 | ~80 |
| Objects365 (Box) | 365 | 29,200 | ~80 |
| COCO (Box) | 80 | 6,400 | ~80 |

In our study, we adopt the Kullback-Leibler divergence to quantify the similarity between two given discrete distributions. Specifically, let $p$ and $q$ denote the estimated probabilistic priors derived from the real-world data and the VISORGPT, respectively. The degree of similarity between these two distributions can be computed as:

$$\text{KL}(p||q) = p\log(p/q). \tag{2}$$

## 6.6 Visualization

**Relation Prior of COCO**. Fig. 9 illustrates the comparison between the real and learned relation prior among 80 categories on the COCO dataset. As can be observed, there is a high degree of similarity between the two relation matrices.

**More Visual Comparison**. We provide more comparison of visual prior between the real world and one learned by our VISORGPT and failure cases on COCO dataset in Fig. 11.

**Continuous Generation**. Fig. 12 presents a set of examples showcasing continuous generation based on the current scene. Notably, in each row, the proposed VISORGPT is able to successfully complete a scene that involves many individuals annotated with 14/18 keypoints or objects with bounding boxes, based on the information provided in the corresponding scene depicted in the previous columns.

Figs. 13 and 14 present more visualization results.

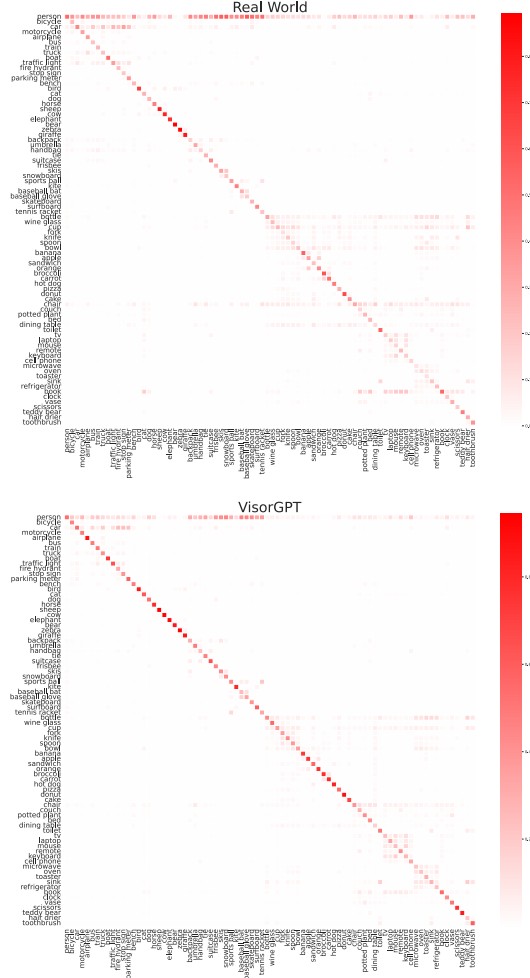

Figure 9: Relation among 80 categories on COCO.

(a) Prior of 'surfboard' adapts to variations of the 'sea'   (b) Supports to receive natural language as input

(c) Scene completion based on user provided objects   (d) Multimodal inference of boxes and keypoints

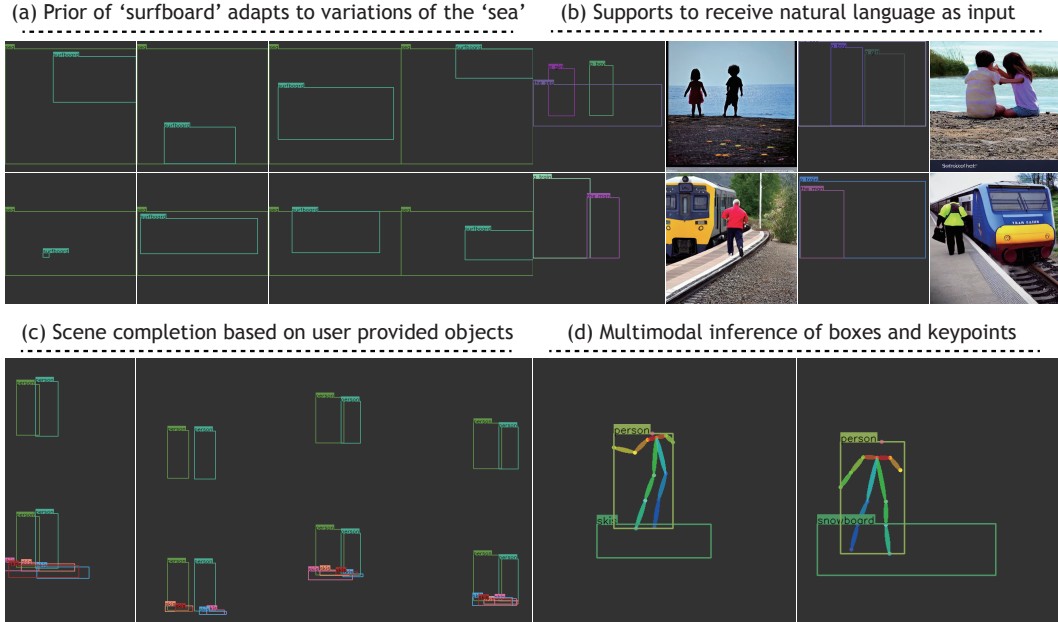

Figure 10: More visual examples.

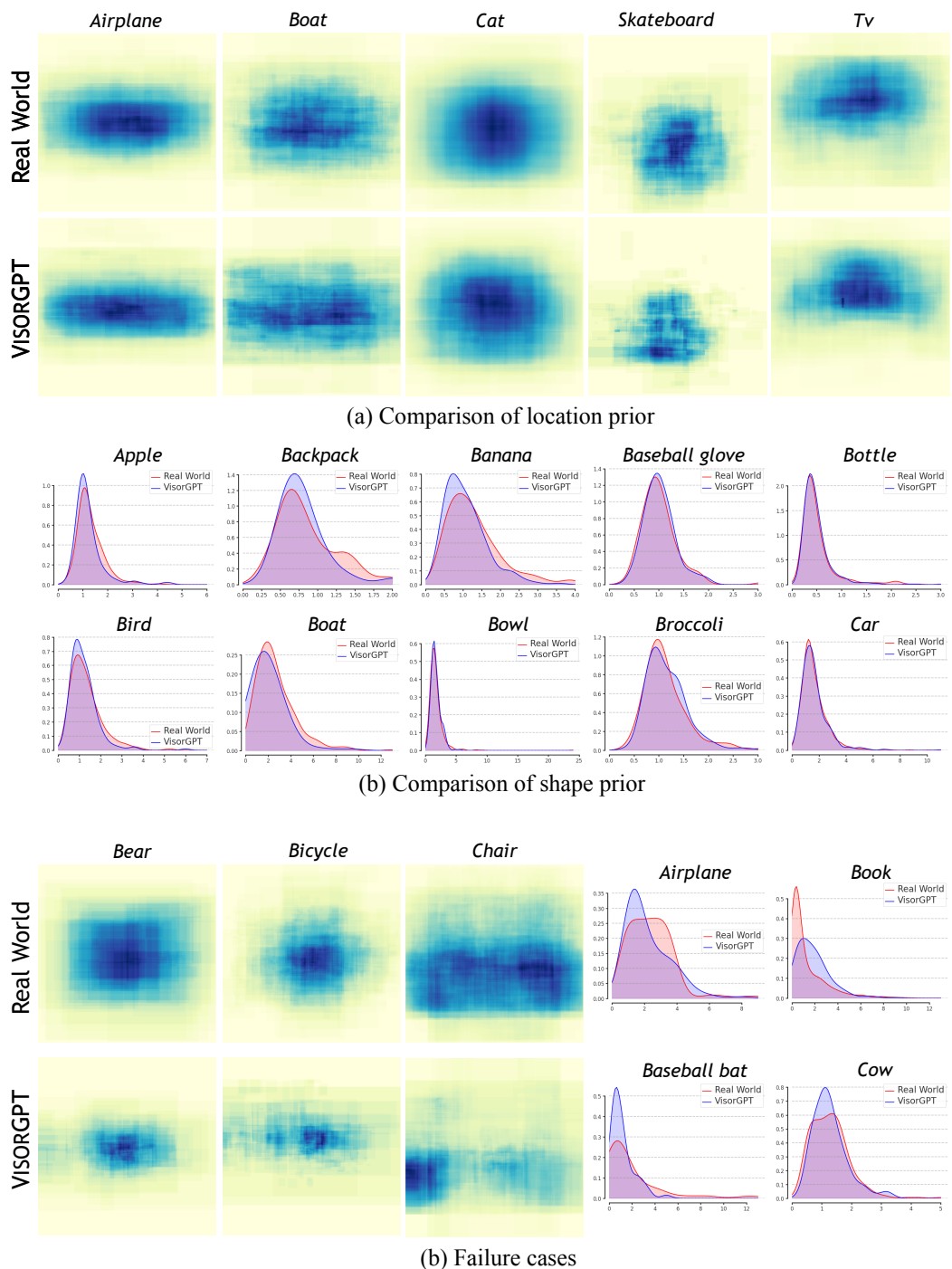

(a) Comparison of location prior

(b) Comparison of shape prior

(b) Failure cases

Figure 11: Comparison of visual prior between the real world and one learned by VISORGPT on COCO dataset.

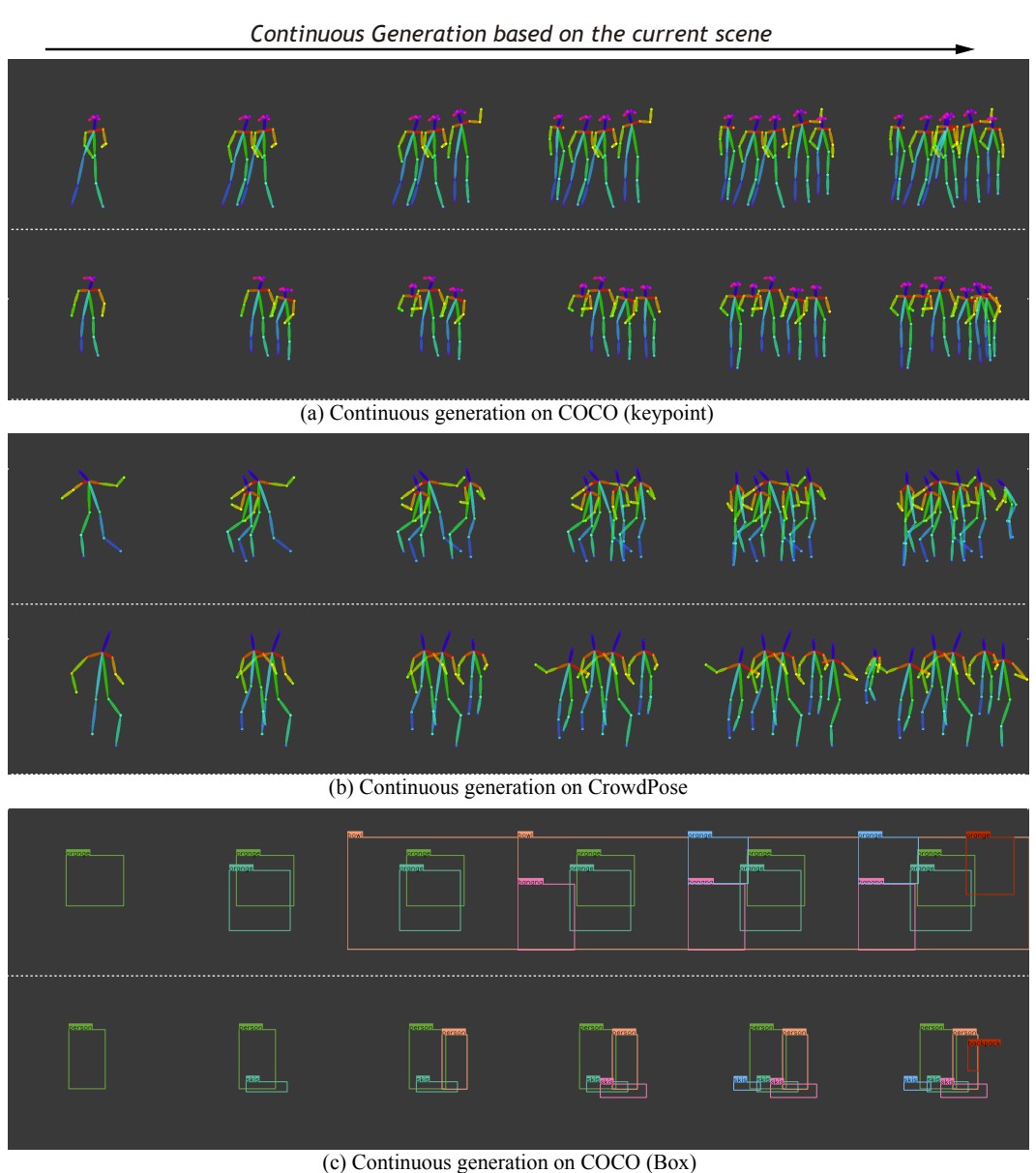

*Continuous Generation based on the current scene*

(a) Continuous generation on COCO (keypoint)

(b) Continuous generation on CrowdPose

(c) Continuous generation on COCO (Box)

Figure 12: Examples of continual generation.

**Input prompt:**

key point; multiple instances; large; 13; 18; person,

**Output sequence:**

key point; multiple instances; large; 13; 18; person, person, person, person, person, person, person, person, person, person, person, person, person; [ a 444 184 b 438 197 c 450 196 d 463 192 e 444 179 f 427 197 g 0 0 h 0 0 i 447 236 j 450 268 k 452 302 l 431 237 m 434 269 n 430 299 o 445 179 p 440 179 q 0 0 r 432 180 ] [ a 232 168 b 223 178 c 238 174 d 267 150 e 256 142 f 208 182 g 183 187 h 149 192 i 230 246 j 262 291 k 281 345 l 204 248 m 208 298 n 187 338 o 236 163 p 228 164 q 0 0 r 217 163 ] [ a 56 139 b 50 154 c 64 149 d 88 176 e 70 163 f 36 159 g 48 193 h 54 166 i 72 215 j 75 255 k 88 293 l 56 218 m 61 265 n 64 302 o 56 134 p 52 134 q 0 0 r 42 136 ] [ a 148 145 b 137 162 c 150 162 d 0 0 e 0 0 f 124 161 g 122 188 h 129 216 i 150 225 j 0 0 k 0 0 l 132 228 m 145 264 n 0 0 o 150 142 p 142 143 q 0 0 r 132 141 ] [ a 352 177 b 342 188 c 358 186 d 364 209 e 376 224 f 326 190 g 316 217 h 341 235 i 358 243 j 362 280 k 364 324 l 335 248 m 338 284 n 334 322 o 354 173 p 348 174 q 0 0 r 336 174 ] [ a 0 0 b 248 31 c 13 136 d 28 196 e 0 0 f 0 0 g 0 0 h 0 0 i 1 233 j 5 316 k 7 383 l 0 0 m 0 0 n 0 0 o 0 0 p 0 0 q 0 0 r 0 0 ] [ a 452 145 b 462 158 c 480 157 d 495 132 e 484 126 f 444 159 g 422 138 h 423 116 i 462 217 j 462 254 k 484 308 l 440 217 m 440 253 n 436 305 o 457 139 p 449 139 q 466 137 r 0 0 ] [ a 204 162 b 201 185 c 219 186 d 0 0 e 0 0 f 183 184 g 178 212 h 197 228 i 217 237 j 0 0 k 0 0 l 192 240 m 0 0 n 0 0 o 208 159 p 201 159 q 212 162 r 194 162 ] [ a 176 170 b 167 181 c 180 184 d 188 212 e 182 192 f 154 178 g 142 209 h 159 196 i 182 249 j 198 280 k 196 325 l 158 250 m 167 298 n 0 0 o 180 168 p 174 165 q 0 0 r 166 163 ] [ a 382 188 b 379 202 c 396 202 d 410 202 e 394 171 f 363 202 g 0 0 h 0 0 i 388 249 j 391 295 k 386 342 l 360 250 m 364 294 n 368 340 o 387 186 p 379 183 q 390 189 r 369 186 ] [ a 284 141 b 276 164 c 301 163 d 324 140 e 331 118 f 252 166 g 220 176 h 190 182 i 289 233 j 316 282 k 303 354 l 260 232 m 243 291 n 212 341 o 288 135 p 277 136 q 0 0 r 267 138 ] [ a 0 0 b 402 170 c 397 172 d 0 0 e 0 0 f 407 169 g 416 197 h 0 0 i 397 215 j 0 0 k 0 0 l 405 213 m 400 257 n 392 289 o 0 0 p 0 0 q 0 0 r 412 156 ] [ a 109 176 b 106 188 c 117 188 d 119 213 e 108 199 f 96 187 g 91 209 h 92 194 i 112 232 j 113 263 k 0 0 l 97 233 m 97 264 n 0 0 o 111 174 p 106 174 q 0 0 r 100 175 ]

**Input prompt:**

key point; multiple instances; large; 8; 18; person,

**Output sequence:**

key point ; multiple instances ; large ; 8 ; 18 ; person , person , person , person , person , person , person , person ; [ a 88 184 b 78 222 c 40 227 d 9 314 e 67 315 f 115 217 g 0 0 h 0 0 i 65 390 j 0 0 k 0 0 l 118 379 m 0 0 n 0 0 o 73 172 p 100 171 q 55 181 r 112 172 ] [ a 452 188 b 222 94 c 447 248 d 416 295 e 0 0 f 0 0 g 0 0 h 474 328 i 438 363 j 0 0 k 0 0 l 492 371 m 0 0 n 0 0 o 441 181 p 465 179 q 431 189 r 487 185 ] [ a 0 0 b 0 0 c 0 0 d 0 0 e 0 0 f 0 0 g 0 0 h 0 0 i 0 0 j 0 0 k 0 0 l 421 268 m 423 302 n 426 325 o 0 0 p 0 0 q 0 0 r 0 0 ] [ a 268 200 b 278 223 c 254 222 d 0 0 e 0 0 f 300 224 g 316 257 h 308 290 i 263 312 j 268 364 k 276 406 l 297 312 m 300 366 n 301 410 o 264 196 p 274 194 q 0 0 r 287 196 ] [ a 168 181 b 166 221 c 133 225 d 121 270 e 160 273 f 201 218 g 228 258 h 236 273 i 147 313 j 165 376 k 171 434 l 188 308 m 188 371 n 196 440 o 160 174 p 175 174 q 148 182 r 184 182 ] [ a 0 0 b 0 0 c 0 0 d 0 0 e 0 0 f 0 0 g 0 0 h 0 0 i 0 0 j 300 287 j 306 340 k 316 388 l 274 280 m 280 338 n 278 391 o 0 0 p 0 0 q 0 0 r 0 0 ] [ a 383 216 b 401 238 c 390 238 d 0 0 e 0 0 f 413 237 g 428 274 h 408 298 i 388 303 j 0 0 k 0 0 l 409 303 m 411 362 n 419 407 o 379 212 p 387 210 q 0 0 r 403 210 ] [ a 219 200 b 220 224 c 194 225 d 0 0 e 0 0 f 246 224 g 0 0 h 0 0 i 196 303 j 202 368 k 203 418 l 238 305 m 238 364 n 237 415 o 214 197 p 224 197 q 208 200 r 0 0 ]

**Input prompt:**

key point; multiple instances; large; 8; 18; person,

**Output sequence:**

key point ; multiple instances ; large ; 8 ; 18 ; person , person , person , person , person , person , person , person ; [ a 88 184 b 78 222 c 40 227 d 9 314 e 67 315 f 115 217 g 0 0 h 0 0 i 65 390 j 0 0 k 0 0 l 118 379 m 0 0 n 0 0 o 73 172 p 100 171 q 55 181 r 112 172 ] [ a 452 188 b 222 94 c 447 248 d 416 295 e 0 0 f 0 0 g 0 0 h 474 328 i 438 363 j 0 0 k 0 0 l 492 371 m 0 0 n 0 0 o 441 181 p 465 179 q 431 189 r 487 185 ] [ a 0 0 b 0 0 c 0 0 d 0 0 e 0 0 f 0 0 g 0 0 h 0 0 i 0 0 j 0 0 k 0 0 l 421 268 m 423 302 n 426 325 o 0 0 p 0 0 q 0 0 r 0 0 ] [ a 268 200 b 278 223 c 254 222 d 0 0 e 0 0 f 300 224 g 316 257 h 308 290 i 263 312 j 268 364 k 276 406 l 297 312 m 300 366 n 301 410 o 264 196 p 274 194 q 0 0 r 287 196 ] [ a 168 181 b 166 221 c 133 225 d 121 270 e 160 273 f 201 218 g 228 258 h 236 273 i 147 313 j 165 376 k 171 434 l 188 308 m 188 371 n 196 440 o 160 174 p 175 174 q 148 182 r 184 182 ] [ a 0 0 b 0 0 c 0 0 d 0 0 e 0 0 f 0 0 g 0 0 h 0 0 i 0 0 j 300 287 j 306 340 k 316 388 l 274 280 m 280 338 n 278 391 o 0 0 p 0 0 q 0 0 r 0 0 ] [ a 383 216 b 401 238 c 390 238 d 0 0 e 0 0 f 413 237 g 428 274 h 408 298 i 388 303 j 0 0 k 0 0 l 409 303 m 411 362 n 419 407 o 379 212 p 387 210 q 0 0 r 403 210 ] [ a 219 200 b 220 224 c 194 225 d 0 0 e 0 0 f 246 224 g 0 0 h 0 0 i 196 303 j 202 368 k 203 418 l 238 305 m 238 364 n 237 415 o 214 197 p 224 197 q 208 200 r 0 0 ]

**Input prompt:**

key point; multiple instances; large; 5; 18; person,

**Output sequence:**

key point ; multiple instances ; medium ; 5 ; 18 ; person , person , person , person , person ; [ a 0 0 b 292 241 c 308 241 d 316 268 e 315 282 f 276 241 g 256 264 h 0 0 i 308 272 j 318 298 k 323 336 l 282 272 m 268 295 n 266 336 o 0 0 p 0 0 q 295 239 r 279 240 ] [ a 0 0 b 236 209 c 251 209 d 262 238 e 270 262 f 221 209 g 202 225 h 199 249 i 248 256 j 249 303 k 0 0 l 224 256 m 217 300 n 0 0 o 0 0 p 0 0 q 247 192 r 236 192 ] [ a 0 0 b 338 247 c 356 248 d 358 273 e 0 0 f 319 246 g 314 272 h 303 289 i 347 299 j 344 332 k 339 372 l 323 297 m 313 328 n 315 366 o 0 0 p 0 0 q 344 226 r 330 226 ] [ a 0 0 b 108 238 c 128 236 d 0 0 e 0 0 f 89 240 g 83 298 h 82 278 i 130 307 j 0 0 k 0 0 l 95 307 m 0 0 n 0 0 o 0 0 p 0 0 q 0 0 r 96 216 ] [ a 0 0 b 406 262 c 424 262 d 433 292 e 433 312 f 388 263 g 378 293 h 381 316 i 418 306 j 428 338 k 440 381 l 396 306 m 384 351 n 388 387 o 0 0 p 0 0 q 414 240 r 398 243 ]

**Input prompt:**

key point; multiple instances; large; 3; 18; person,

**Output sequence:**

key point ; multiple instances ; large ; 3 ; 18 ; person , person , person ; [ a 164 220 b 167 266 c 203 279 d 0 0 e 212 324 f 132 252 g 95 302 h 112 338 i 184 352 j 0 0 k 0 0 l 127 344 m 0 0 n 0 0 o 176 210 p 158 210 q 192 215 r 146 214 ] [ a 378 214 b 407 260 c 444 268 d 456 340 e 393 341 f 370 253 g 367 309 h 360 354 i 427 381 j 0 0 k 0 0 l 372 368 m 0 0 n 0 0 o 390 202 p 370 204 q 414 203 r 0 0 ] [ a 296 238 b 317 281 c 355 289 d 355 349 e 299 364 f 278 273 g 263 324 h 274 353 i 336 391 j 0 0 k 0 0 l 282 382 m 0 0 n 0 0 o 0 0 p 307 232 q 292 230 q 329 237 r 0 0 ]

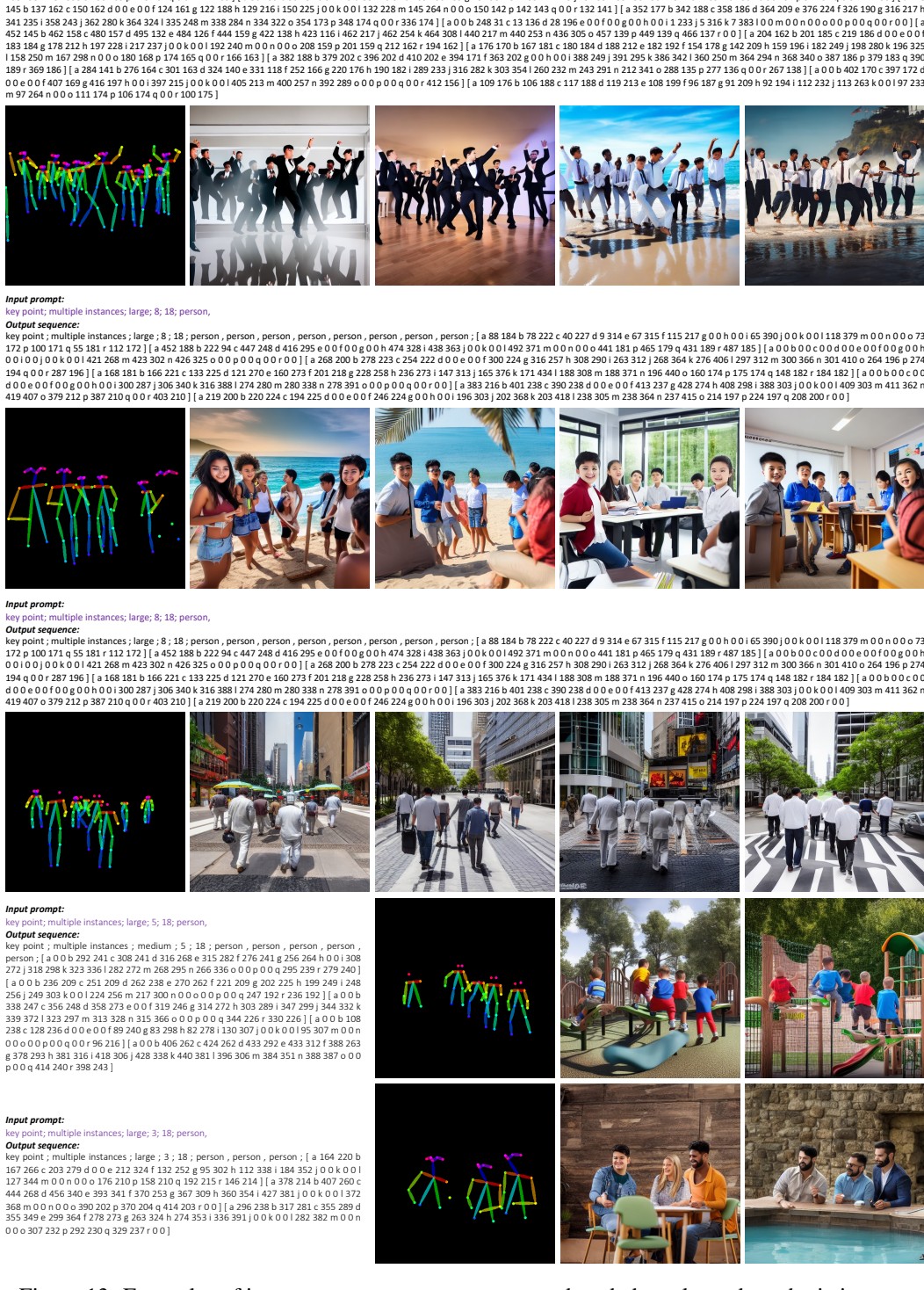

Figure 13: Examples of input prompts, output sequences, decoded results and synthetic images.

**Input prompt:**
box; multiple instances; large; 5; 0; dining table, knife, apple, apple, apple;
**Output sequence:**
box; multiple instances; large; 5; 0; dining table, knife, apple, apple, apple; [ xmin 1 ymin 86 xmax 512 ymax 421 ] [ xmin 0 ymin 85 xmax 100 ymax 388 ] [ xmin 203 ymin 191 xmax 344 ymax 299 ] [ xmin 366 ymin 254 xmax 472 ymax 355 ] [ xmin 194 ymin 203 xmax 241 ymax 274 ]

**Input prompt:**
box; multiple instances; large; 5; 0; horse, person, person, horse, person;
**Output sequence:**
box; multiple instances; large; 5; 0; horse, person, person, horse, person; [ xmin 1 ymin 164 xmax 219 ymax 414 ] [ xmin 362 ymin 198 xmax 442 ymax 396 ] [ xmin 234 ymin 169 xmax 316 ymax 327 ] [ xmin 305 ymin 190 xmax 475 ymax 401 ] [ xmin 350 ymin 203 xmax 383 ymax 297 ]

**Input prompt:**
box; multiple instances; large; 5; 0; horse, horse, horse, horse, horse;
**Output sequence:**
box; multiple instances; large; 5; 0; horse , horse , horse , horse , horse; [ xmin 0 ymin 212 xmax 97 ymax 267 ] [ xmin 0 ymin 166 xmax 59 ymax 237 ] [ xmin 89 ymin 175 xmax 180 ymax 225 ] [ xmin 235 ymin 144 xmax 512 ymax 421 ] [ xmin 205 ymin 215 xmax 268 ymax 238 ]

**Input prompt:**
box; multiple instances; large; 7; 0; person, cup, donut, donut, donut, donut, dining table;
**Output sequence:**
box; multiple instances; large; 7; 0; person, cup, donut, donut, donut, donut, dining table; [ xmin 0 ymin 64 xmax 509 ymax 158 ] [ xmin 37 ymin 136 xmax 183 ymax 316 ] [ xmin 235 ymin 144 xmax 352 ymax 249 ] [ xmin 339 ymin 243 xmax 479 ymax 386 ] [ xmin 212 ymin 239 xmax 357 ymax 378 ] [ xmin 354 ymin 198 xmax 485 ymax 296 ] [ xmin 1 ymin 64 xmax 511 ymax 442 ]

**Input prompt:**
box; multiple instances; large; 5; 0; boat, person, person, umbrella, tie;
**Output sequence:**
box; multiple instances; large; 5; 0; boat, person, person, umbrella, tie; [ xmin 0 ymin 254 xmax 511 ymax 443 ] [ xmin 245 ymin 146 xmax 381 ymax 391 ] [ xmin 77 ymin 136 xmax 243 ymax 376 ] [ xmin 340 ymin 98 xmax 510 ymax 193 ] [ xmin 176 ymin 211 xmax 199 ymax 250 ]

**Input prompt:**
box; multiple instances; large; 5; 0; vase, chair, dining table, vase, chair;
**Output sequence:**
box; multiple instances; large; 5; 0; vase, chair, dining table, vase, chair; [ xmin 189 ymin 249 xmax 305 ymax 411 ] [ xmin 85 ymin 251 xmax 156 ymax 393 ] [ xmin 85 ymin 347 xmax 423 ymax 506 ] [ xmin 293 ymin 276 xmax 377 ymax 417 ] [ xmin 345 ymin 243 xmax 427 ymax 346 ]

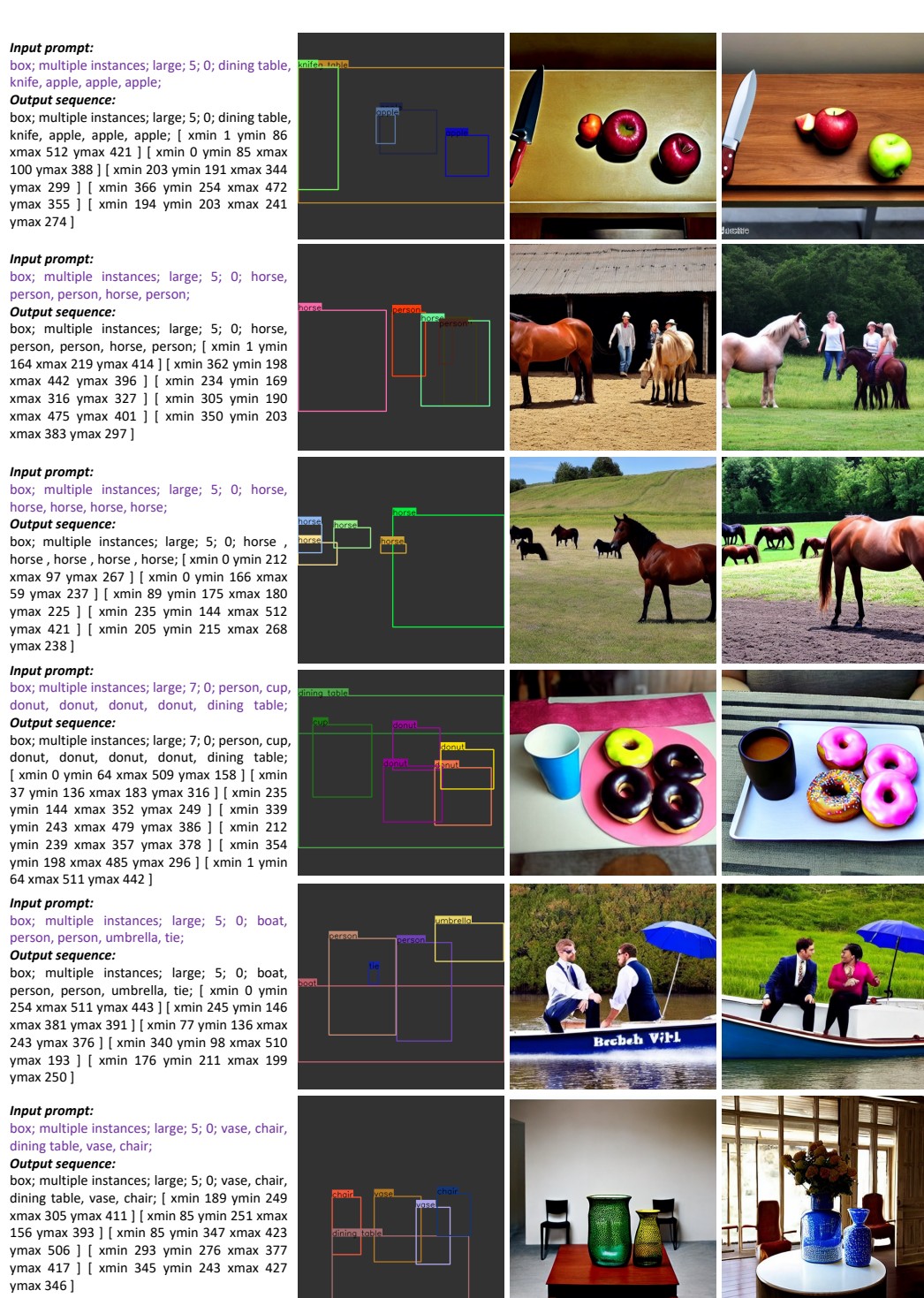

Figure 14: Examples of input prompts, output sequences, decoded results, and synthetic images.

