# OpenReview forum: "Learning Visual Prior via Generative Pre-Training"
_NeurIPS.cc/2023/Conference — NeurIPS 2023 poster_

### Official Review · Reviewer_CQCW · 2023-06-13

**Soundness:** 3 good
**Presentation:** 3 good
**Contribution:** 2 fair
**Rating:** 6
**Confidence:** 3

**Summary:**

The paper proposes VisorGPT to model the propabilistic visual prior via generative pretraining. This is achieved by transforming the continuous values (locations, bounding boxes) into discrete tokens. Prompt engineering are employed to arrange these tokens into a sequence. The author conducts experiments to prove that VisorGPT is able to model the visual prior effectively and boost various downstream tasks like conditional image generation.

**Strengths:**

(1) The paper proposes a novel setting: modeling visual prior via generative pretraining. This could be a potential area to combine visual prior with the power of large language models.

(2) The paper is easy to follow, and includes comprehensive quantitative experiments and good visualization.

**Weaknesses:**

(1) The rationale for utilizing GPT to model visual priors is not well-explained. Given that many visual priors, such as bounding boxes and point locations, are continuous in nature, a straightforward baseline could be an autoregressive model with a regression head. It would be beneficial to provide comparisons with baseline methods and justification for using GPT in visual prior modeling.

(2) The controllability of the proposed VisorGPT method appears to be overstated. While it can manipulate the number, size, and type of objects, the user is unable to control the position of each generated object or their relative positions. Furthermore, existing research (e.g., LayoutGPT [1]) has attempted to control the number, size, and type of objects using language models without additional training. As a result, the impact of VisorGPT may be somewhat limited.

[1] LayoutGPT: Compositional Visual Planning and Generation with Large Language Models

**Questions:**

(1) As stated in the weakness session, I am curious about the motivation to model the visual prior via GPT.

(2) In experiment setting, the authors use KL divergence as a metric. However, I cannot figure out how to calculate the KL score, like how to estimate the distribution over the real bounding boxes?



**Limitations:**

The author has discussed about the limitations of this work.

---

> ### Author Rebuttal · Authors · 2023-08-10
>
> **The Figures and Tables are provided in the global pdf**
> 1. The rationale and motivation for utilizing GPT to model visual priors.
> There are three main considerations:
> i) **Visual priors are not just location**. Our visual priors scope extends beyond the modeling of pure continuous numbers (e.g., location). Instead, we aim to unify locations, categories, and texts into a sequence representation, enabling a straightforward learning process without the need to design additional modules to adapt/process different types of modalities of information.
> ii) **Textual semantic understanding**. With the capacity to model locations, categories, and texts together, the model can learn synonyms of categories and relationships among different categories from the texts, enabling the extrapolation of synonyms or new categories based on the acquired priors of bounding boxes. Hence, the model is capable of addressing open-vocabulary scenarios. We present examples in Fig. 5, where 'girl,' 'boy,' and 'man' serve as synonyms for 'person.' These results will be included in the paper.
> iii) **Fulfill requirements by prompting and completing**. The Generative Pre-Training (GPT) paradigm facilitates flexible and personalized applications through prompting. For instance, by inputting a sequence like 'box;' or 'keypoint;' to VisorGPT, it can generate corresponding box or keypoint information. Additionally, as depicted in Fig. 3(a), we can provide initial layouts ('person, person') for VisorGPT, which then infers the missing classes along with their bounding boxes (Fig. 3(b)). These functionalities are attributed to the learning paradigm, i.e., modeling all modalities within a sequence.
> 2. Comparisons to baseline methods.
> To enhance the evaluation, we reproduce recent SOTA methods such as LayoutDM(CVPR’23) [1] for designing the layout of documents and mobile applications, and VQDiffusion(CVPR’22) [2] on COCO dataset for comparison. We also apply VisorGPT to documents and mobile applications designing benchmarks like Rico [3] and PubLayNet [4]. As shown in Table 1, the experimental results include FID and Align. score (referred from LayoutDM) with lower values indicates better performance. Notably, **VisorGPT significantly surpasses these baseline and SOTA methods on all three datasets**, showcasing the superior capability of VisorGPT to learn layout generation. We will include these results in the paper.
> 3. The controllability of VisorGPT.
> Apart from controlling the number, size, and type of objects, VisorGPT can also control the position of objects in three aspects:
> i) As shown in Fig. 3, **users can control some objects’ positions by first providing their bounding boxes**, and the VisorGPT can receive these information to infer a complete layouts. In Fig. 3(a), when the objective is to infer six instances ('person, person, skis, skis, skis, skis'), users can provide positions of two instances ('person, person') and VisorGPT can then generates the remaining four instances (Fig. 3(b)), with consistency to the user-provided ones.
> ii) Besides, **VisorGPT currently can receive natural language input**. Hence, users can control the relative positions among objects by using natural language. As shown in Fig. 5, given input text prompt “A boy and a girl are playing by the sea.”, VisorGPT can output corresponding bounding boxes adhering to such relative positions.
> iii) Further, we have extended VisorGPT to receive users’ provided coarse layouts as input and refine such coarse layouts to a more reasonable ones based on the learned priors. Hence, users can potentially control each object’s position.
> These results will be included in the latest version of paper.
> 4. Comparison to the concurrent work LayoutGPT.
> i) LayoutGPT reasons layout by textual knowledge and in-context examples. When encountering very complex scenes involving a large number of objects or segmentation masks, LLM textual knowledge is challenging to understand visual layout from limited examples with dense numerical data. In contrast, VisorGPT, with large-scale pretraining on natural layout corpus, can generate not only numerous objects (as depicted on the right in Fig. 4) but also segmentation masks.
> ii) Compared to LayoutGPT which generates the layout by starting from fix-style input), training-based VisorGPT can complete the layout based on in-consistent conditions (e.g., starting by category, numerical box, or key points), which offers greater personalization potential for condition generation. For instance, as shown in Fig. 3, the Tb prompt format (introduced in section 3.2) empowers VisorGPT to perform scene completion based on users' provided pre-defined layouts.
> iii) **LayoutGPT relies on the paid GPT-3.5/4 API, incurring costs for users**. On the other hand, based on publicly available GPT-2, VisorGPT will be freely accessible.
> We will incorporate these relevant discussions about LayoutGPT in the paper.
> 5. How to calculate the KL DIV metric?
> For example, to estimate the shape prior of a category such as 'dog' we calculate the ratio of width to height of all bounding boxes associated with dogs in the test set, generating a list of numerical results that we use to construct a histogram representing the estimated distribution. Similarly, for VisorGPT, we can employ the same method to estimate a distribution, and subsequently calculate the KL divergence between real-world  and VisorGPT distributions.
> Additionally, we enhance our evaluation by incorporating FID and Align scores from relevant literature on document and mobile application design (Table 1). The results will be included in the paper.
>
> [1] Layoutdm: discrete diffusion model for controllable layout generation
> [2] Vector Quantized Diffusion Model for Text-to-Image Synthesis
> [3] Rico: A Mobile App Dataset for Building Data-Driven Design Applications
> [4] PubLayNet: largest dataset ever for document layout analysis

---

> > ### Comment · Reviewer_CQCW · 2023-08-10
> > **Additional question about controllability**
> >
> > Thanks for the clarification and additional results. They have addressed most of my concerns. I do have a follow-up question regarding the controllability of your system. Can your system facilitate iterative fine-tuning by the user? For instance, if the system initially generates a layout based on the prompt "A boy and a girl are playing by the sea," would it be possible for the user to subsequently modify the output by prompting, "I want to make them closer" or "I want the boy to be taller"?
> >
> > Also, do you think it would be possible, or do you have plan, to extend this system to infer layout in 3D space?

---

> > > ### Author Response · Authors · 2023-08-11
> > > **Reply to the additional question about controllability**
> > >
> > > Thank you for your prompt reply. We conducted the suggested experiments and discovered that **VisorGPT can effectively facilitate the iterative fine-tuning of target layouts through natural language prompts**, such as "I want to make them closer," "I want the boy to be taller," "I want the boy to be xx pixels taller," or other requests while maintaining the positions of non-target objects unchanged. To achieve this functionality, we introduced input-instruction-output pairs like [original layouts]-[instructions like “I want {} to be taller”]-[modified layouts] into the training process of VisorGPT. Importantly, the creation of such pairs is fully automated. After training, VisorGPT can be used to interactively refine or modify provided layouts using natural language. We will integrate this function into VisorGPT and provide comprehensive details in the paper. We appreciate your valuable suggestion.
> > >
> > > Thank you for pointing out. As presented in our response, our training-based VisorGPT can be tailored to a broad spectrum of applications through prompting. Hence, **it can be extended to 3D space such as designing the indoor scene layouts**, and other domains, such as video space. We are working to enhance VisorGPT's utility by extending it to these areas and will add the corresponding results, further enriching its versatility and potential value to the research community.

---

> > > > ### Comment · Reviewer_CQCW · 2023-08-12
> > > > **Reply to Reply to the additional question about controllability**
> > > >
> > > > I appreciate the clarification provided. I've noticed that input-instruction-output pairs are incorporated in the model training process. I have a few questions. Firstly, could you please specify whether the model is trained from scratch or if it undergoes a fine-tuning process with such new data? Additionally, I am curious to learn about the time and resource requirements for training the model. It would be beneficial for the community if the training time and GPUs used were included in the model card table. This information would provide valuable insights to users.

---

> > > > > ### Author Response · Authors · 2023-08-13
> > > > > **Reply to questions about the time and resource requirements**
> > > > >
> > > > > For the sake of efficiency, we use the LoRA technique to fine-tune the model on the input-instruction-output pairs. Note that we also leverage DeepSpeed to reduce the memory consumption.
> > > > >
> > > > > **We sincerely appreciate your valuable suggestions aimed at improving the clarity of our paper**. Here, we provide the training duration and resource requirements of VisorGPT as outlined below:
> > > > > | Model |Training type |GPU resources |Training time |
> > > > > | :---        | :----: |          :---: |         :---: |
> > > > > |VisorGPT|train from scratch (in paper)| 8xV100 | ~56 hours|
> > > > > |VisorGPT|instruction tuning (this time) |4xV100|~6 hours|
> > > > >
> > > > > We will incorporate the details into the model card in the paper.
> > > > >
> > > > > To enhance user convenience when using VisorGPT, **we will integrate these functions into a unified VisorGPT model, which will be made accessible to the community**. Additionally, we will release the **full training/fine-tuning code and scripts**, and we are actively working on implementing additional strategies within DeepSpeed to **further reduce memory consumption**. This will facilate users to personalize and explore new functions of VisorGPT with greater ease.

---

> > > > > > ### Comment · Reviewer_CQCW · 2023-08-14
> > > > > > **Reply**
> > > > > >
> > > > > > Thanks. My questions were addressed adequately. I have raised my rating to positive.

---

> > > > > > > ### Author Response · Authors · 2023-08-15
> > > > > > > **Reply**
> > > > > > >
> > > > > > > Thank you for your valuable suggestions to enhance our paper and raise the rating. We will include these experimental results in the latest verion. Your dedication to participating in meaningful discussions is greatly appreciated.

---

### Official Review · Reviewer_xtC6 · 2023-06-23

**Soundness:** 2 fair
**Presentation:** 2 fair
**Contribution:** 2 fair
**Rating:** 6
**Confidence:** 4

**Summary:**

This paper proposes to translate image layout information such as bounding box layouts, masks and keypoints into discrete sequences that can be modelled by NLP architectures, such as GPT. By doing so, the image layout distribution of a dataset can be learned to later sample from it and obtain plausible in-distribution layouts. The experimental section shows that GPT learns to sequentially model these layouts successfully, and that the layouts can be later used as input to bounding box to image methods.

**Strengths:**

- The method that this paper proposes to translate layouts into sequences is reasonable.
- There has been efforts towards the reproducibility of the paper, although some details are missing.
- The experimental section shows that the trained GPT model is successfully modelling the dataset distribution.
- The output BB layouts and keypoints were used as input to generative models to obtain plausible generated images.

**Weaknesses:**

- The motivation and significance of this work is not clear and needs more work to ground it with a specific application.
    - Why is learning the layout distribution of a dataset important? This idea works for in-distribution, but how is it useful when we want to generalize to unlikely or unseen layouts? With a creative user in mind, it is more than likely that unseen conditionings will be used. To enable controllability, generalization becomes really important. If the plan is to restrict conditionings to the in-distribution layouts, then controllability is sacrificed. Then, what is the end-goal of this work?
    - Learning the distribution of the layouts is useful to sample conditionings that are very similar to the ground-truth (GT) BB layouts in the dataset. However, how is it different than filtering ground-truth BB layouts by specifying how many instances/sizes/category names we want? It would be interesting to see how the BB layouts and generated images compare when using the proposed method versus sampling GT layouts, across several samples given the same conditioning. Does the model provide more diversity and richer BB layouts than the GT ones? What is the real advantage of the model?
    - The input prompts are not very user friendly and steer away from natural language.


- The evaluation section has significant room for improvement.
    - It lacks comparison with any baseline.
    - It lacks quantitative image quality/diversity/controllability comparison with other ways of obtaining bounding boxes and compared to text prompts. The most naive and simple baseline to obtain BB layouts is to use the GT ones.
    - The paper presents several ways of generating image layouts (BB, masks, keypoints) but only BB layouts are evaluated.
    - In Table 4, why using Ta + Tb instead of Tb alone?




- The writing lacks clarity. Many details with respect to the evaluation are unclear or not properly conveyed in the text:
    - l.59 "...generalized visual intelligence models". As mentioned above regarding the restriction of controllability, having this model output the BB layouts goes in the opposite direction of generalization.
    - Section 3.1. I found it confusing to call the conditionings x, as they often referred to as y. The x variable is usually employed when talking about images.
    - Figure 2. Given the examples of training sequences in Appendix 1.1, it is unclear how the instructions "Add 2 person" are learned. Lacking reproducibility details.
    - Table 2. Unclear what "sampling prop." and "epochs" mean here. Was the model trained on each dataset sequentially? Given the comments regarding the dataset balancing I do not believe it is the case, but this table induces confusion.
    - Evaluation metrics. "Format: whether it can be successfully decoded" how is this measured?. "Matching: number of categories matches the number of locations" what does this mean?
    - It seems "shape prior" is the same as "aspect ratio". Consider changing for clarity.
    - Table 5. "Quality" and "Controllability" in this table could be confused by image quality and controllability, while this table is about matching statistics with the GT BB layouts.
    - Table 7. Lack of context. What is SW, TK and #Seq in these experiments?
    - How were the "not obeying prior" conditionings in Figure 5 obtained?

**Questions:**

Questions were asked above in the weaknesses section.

Overall, one way of steering this work towards a useful application would be to use user text prompts, written as natural language sentences, and to show that they can be changed towards more in-distribution BB layouts to obtain better image quality, given the learned prior, with a consistent and quantitative evaluation. This would be a trade-off in controllability/image quality.

**Limitations:**

The paper addresses the limitations and societal impact.

---

> ### Author Rebuttal · Authors · 2023-08-10
>
> **The Figures and Tables are provided in the global pdf**
> 1. Generalize to novel layouts.
> The training set to train VisorGPT contains approximately 4 million images with around 40 million bounding boxes, providing **an extensive and diverse range of layouts to enable novel layout generation**. We utilize a set of target classes ('bottle', 'dining table', 'person', 'knife', 'bowl', 'bowl', 'oven', 'person', 'cup', 'cup', 'bowl', 'bowl', 'broccoli', 'spoon') to select the relevant layouts from the training set. We observe that only one satisfied sample over 110K samples contains all the target classes (as shown on the left in Fig. 4(a)).
> Subsequently, we employ the same set of target classes as input for VisorGPT and infer their corresponding bounding boxes (as depicted on the right in Fig. 4(a)). Evidently, **the layouts generated by VisorGPT are diverse and differ significantly from the relevant layout selected from the training set**. This validates that VisorGPT does not rely on memorization. Instead, it learns intrinsic visual priors among categories, empowering it to infer novel/unseen layouts.
> Moreover, **VisorGPT can receive user conditions, and reasonably complete the corresponding missing classes in an interactive way**. As depicted in Fig. 3(a), users can roughly draw two instances of 'person' across the canvas and prompt VisorGPT to deduce the remaining 'skis'. As demonstrated in Fig. 3(b), VisorGPT successfully infers the correct positions of four 'skis'. This validates that our **VisorGPT possesses robust generalization capabilities to accommodate the diverse requirements of users**.
> 2. Avantages and diversity comparison to selecting GT bounding boxes from training set.
> (i) **Diverse layouts generation**. When users require to specify a large number of classes (larger than 10), the layouts selected from training set including all such classes are limited. As shown on the left in Fig. 4(a), when target classes ('bottle', 'dining table', 'person', 'knife', 'bowl', 'bowl', 'oven', 'person', 'cup', 'cup', 'bowl', 'bowl', 'broccoli', 'spoon') are considered, there is only one relavant layout containing all the target classes from the training set. Another example is shown in Fig. 4(b). In contrast, as shown on the right of Fig. 4, VisorGPT can receive these target classes to infer a lot of corresponding layouts with much diversity.
> (ii) **User-customized/Controllability**. Compared to selecting from ground truth bounding boxes, one of the main advantages of VisorGPT lies in its ability to extrapolate novel layouts based on users' configurations, as depicted in Fig. 3. VisorGPT can deduce the missing classes’ positions (‘skis, skis, skis, skis’) based on the layout (‘person, person’) given by users.
> (iii) **Free-form captions as layout condition**. Incorporating captions in the training sequences enables VisorGPT to receive free-form natural language input as condition (e.g.,A boy and a girl are playing by the sea) and generate corresponding layouts (as shown in Fig. 5). We will include these results in the paper.
> 3. Natural language input.
> We have extended VisorGPT to accommodate natural language input. As depicted in Fig. 5, when provided with the prompt "a girl and a boy are playing by the sea" VisorGPT can accurately infer the individual and background along with their respective bounding boxes. We will incorporate these results in the paper.
> 4. Comparison to baseline.
> Thanks for your suggestion. To enhance the evaluation, we reproduce recent SOTA methods such as LayoutDM(CVPR’23)[1] for designing layout of documents and mobile applications and VQDiffusion(CVPR’22)[2] on the COCO dataset for comparison. We also apply VisorGPT to documents and mobile applications designing benchmarks like Rico[3] and PubLayNet[4].The results are presented in Table 1(global pdf), which include FID and Align. score (referred from LayoutDM) with lower values indicates better performance. Notably, **VisorGPT significantly surpasses these methods on all three datasets**. We will include these results and more evaluations (mask and pose) in the paper.
> 5. Image quality/controllability comparison to other ways of box generation like the selection of gt ones.
> As suggested, we compare the baseline, which involves using GT boxes to synthesize images, in Table 2. Specifically, we utilize the bounding boxes from the COCO val set as spatial conditions for GLIGEN to synthesize images and use the bounding boxes generated by VisorGPT for GLIGEN to synthesize images. Then, we calculate both the FID and YOLO score [5]. As shown in Table 2, the FID and YOLO score of images produced using the bounding boxes generated by VisorGPT are similar to those obtained using GT boxes. This demonstrates that our generated bounding boxes closely adhere to the real distribution.
> Table 2. Image comparison using GT Bboxes.
> | Method|FID $\downarrow$ |YOLO score (AP)$\uparrow$|
> |:-:|:-:|:-:|
> |GT Bbox |25.62|27.4|
> |VisorGPT|26.37| 26.2|
>
> 6. Why use Ta + Tb instead of Tb alone?
> For users, Ta is simpler and more versatile than Tb, and Tb is specifically designed for scene completion (as shown in Fig. 3). For clear comparison, we will include results of using Tb alone.
>
> 7. Clarity
> Due to the limited maximum length, we cannot answer all the questions. We construct a large dataset to train VisorGPT, and the training is not sequential. We sincerely thank the reviewer for pointing out this lack of clarity, and we will add more descriptions and use clearer language to enhance the clarity of the paper.
>
> [1] Layoutdm: discrete diffusion model for controllable layout generation
> [2] Vector Quantized Diffusion Model for Text-to-Image Synthesis
> [3] Rico: A Mobile App Dataset for Building Data-Driven Design Applications
> [4] PubLayNet: largest dataset ever for document layout analysis
> [5] Image synthesis from layout with localityaware mask adaption

---

> ### Author Response · Authors · 2023-08-20
> **Reply to the clarity**
>
> 1. The term "generalized visual intelligence models".  As demonstrated in our responses, VisorGPT can generate diverse layouts and enable users to control object positions by i) providing initial layouts and ii) natural language input. As shown in the discussion with reviewer [CQCW](https://openreview.net/forum?id=LUT4b9gOtS&noteId=FHUTN9ZDkG), users can ask VisorGPT to further modify the current layouts by natural language such as “I want the boy to be taller”. The above demonstrates our VisorGPT’s good controllability.  Besides, the term “generalized visual intelligence models” refers to VisorGPT's capacity to continuously and autonomously generate spatial layouts for image synthesis models, enabling the synthesis of numerous pairs of images with box/keypoint/mask annotations. Consequently, this approach can potentially yield a substantial increase in data, which in turn enhances existing visual models such as object detection and human pose estimation models to a more generalized ones. We will give more description about this term for clarity.
> 2. Variable x. Thanks for your suggestion. We will replace x with y.
> 3. Instruction of ‘add 2 person’. Specifically, we construct many instruction-input-output pairs to train VisorGPT. Hence, in inference, given the concatenation of the instruction ‘add 2 person’ and the input sequence, VisorGPT can continually infer two people’s keypoints based on the input sequence. We will add these details to the paper.
> 4. Training Details. The training process is non-sequential. We construct a large training set by sampling sequences from each dataset based on the specified sampling proportion (sampling prop.) and subsequently employ it to train VisorGPT. The "epochs" column indicates the estimated number of epochs that will be trained for each dataset. Table 2 in the paper is referenced from Table 1 in LLaMa[1]. To ensure clarity and comprehension, we will provide additional descriptions in the paper.
> 5. Evaluation metric - Format accuracy. It is important to note that the successful decoding of generated sequences into categories and their corresponding bounding boxes relies on the prompt format designed in Section 3.2. To assess the decode success rate of the generated sequences, we propose using "format accuracy." Specifically, we will infer a set of sequences (e.g., 48,000 sequences for Open Image dataset) and then calculate the number of sequences that can be successfully decoded into category names and corresponding boxes, which allows us to compute the format accuracy of the generated sequences. We will include more details in the paper.
> 6. Evaluation metric - Matching. Note that the generated sequences are expected to strictly produce n categories along with their n corresponding bounding boxes. However, there might be cases where the generated sequences contain n categories but with m bounding boxes. To evaluate this aspect, we compute the matching accuracy, which indicates how effectively VisorGPT can infer n objects with their corresponding n boxes. We will include more descriptions or replace it with another word for clarity.
> 7. Change shape prior to aspect ratio. Thanks for your suggestion. We will change it.
> 8. Change quality and controllability. Thanks for your suggestion. We will change other words instead of quality and controllability to avoid confusion.
> 9. SW (Special Words) refers to the special tokens such as 'a, b, c, d' and 'm1, m2, m3' introduced in Section 3.2. To provide a more comprehensive understanding, we will include additional statements about SW in the main body of the paper.
> 10. TK (Textual Knowledge) denotes the pre-trained weights of GPT-2, and #Seq represents the number of generated sequences utilized to calculate format and matching accuracy, as explained in the appendix. For improved clarity, we will incorporate more contextual details into the main body of the paper. Thank you for your valuable suggestion.
> 11. How to obtain the box not adhering to priors. It is obtained from users. Specifically, we remove some layouts from the generated ones and request users to draw the missing categories randomly. We will introduce such a procedure and include more visual examples.
>
> **Thanks for your valuable reviews and suggestions. We will include more corresponding details and descriptions in the final paper to further enhance the clarity.**
>
> [1] LLaMA: Open and Efficient Foundation Language Models.

---

> > ### Comment · Reviewer_xtC6 · 2023-08-20
> > **Acknowledgement**
> >
> > Thank you for the rebuttal and clarifying most of my concerns. I am raising the score accordingly.
> > In general, I am not seeing any of the relevant feedback included in the main paper nor the supplementary, which is much rather preferred than the promise to include it later on.

---

> > > ### Author Response · Authors · 2023-08-21
> > > **Reply**
> > >
> > > **Thank you for your kind feedback and for raising the score**. As the paper and supplementary materials cannot be updated or modified during the rebuttal phase (only one page pdf including figures and tables is allowed), we refrained from making changes at this time. To further enhance the quality of our paper and enrich its potential value to the research community, we promise that we are committed and pleased to incorporate the suggested content and corresponding results into the final version of the paper.

---

### Official Review · Reviewer_6eEZ · 2023-07-03

**Soundness:** 4 excellent
**Presentation:** 4 excellent
**Contribution:** 3 good
**Rating:** 7
**Confidence:** 4

**Summary:**

This is impressive work, striving to explicitly learn visual priors and enabling customization in the sampling process for conditional image synthesis. Such a task normally occurs inaccurate synthetic results due to improper conditional priors, such as spatial location, shape, and interaction of objects.

Taking inspiration from the great success of generative language modeling that estimates the likelihood of a given sequence of words occurring in a sentence, the authors propose VisorGPT which converts visual priors into a corpus of sequences and casts the learning objective as maximizing the likelihood of each sequence. Two universal prompts are proposed to encode different types of visual locations, including object bounding box, human pose, and instance mask.

Owing to the robust modeling capacity of generative pre-training, VisorGPT is successful in effectively modeling visual priors. This allows for the customization of sequential outputs through advanced prompt engineering.


**Strengths:**

- **Scope and relevance**: The topic is very interesting and important. Considering the growing interest in AIGC, this paper fills the blank of learning visual priors for conditional image generation.

- **Significance of contributions**:  The proposed model VisorGPT is novel and has expanded the territory of generative pre-training to learn visual priors, which mitigates the issue of lacking good prior in conditional image synthesis.

- **Experiments**: The authors conduct extensive experiments and demonstrate improved generative results on conditional image generation.

- **Clarity**:  The paper is well-written and easy to follow.

**Weaknesses:**

- **Closed-set bias**: As acknowledged by the authors, the overall categories are approximately 1000. Thus, the proposed VisorGPT may be biased toward closed-set training data. While I acknowledge that this is a pioneering attempt to learn visual priors with generative pre-training, I would recommend that the authors take care to provide additional guidance and precautions for the community in the paper.

- **Data scalability**: The training data of VisorGPT is built upon existing labeled vision datasets. However, labeling these datasets with bounding boxes, instance masks, and key points is very time-consuming and expensive. Therefore, scaling up the training data is an important issue.

- **Limited application**: The authors only focus the application of VisorGPT on conditional image generation. Are there any other potential applications that can be explored to show the usefulness of VisorGPT?

**Questions:**

In addition to the above weaknesses, here are two more questions:

- Despite VisorGPT being interesting and novel, why bother to use generative sequential pre-training to learn visual prior rather than other learning paradigms, such as GAN or VAE? Given that baseline works (may not exist) are not provided for comparison, what are the key advantages of VisorGPT?

- Can we use generated images with learned prior as labeled data for training models for object detection, instance segmentation, and keypoint detection tasks?

**Limitations:**

The authors haven't adequately addressed the potential negative societal impact of their work.

---

> ### Author Rebuttal · Authors · 2023-08-10
>
> **The Figures and Tables are provided in the global pdf.**
> 1. Add guidance and precaution to address the closed-set bias.
>     Thanks for your valuable suggestion. Actually, 1,000 categories learned by VisorGPT can cover most of the common objects in the world. To include more potential categories, here we give some potential guidance. As we know, the large-scale text corpus contains much knowledge about the world and also the relation among different categories. Hence, if the text corpus is incorporated to train VisorGPT, the bounding boxes of unknown objects can be inferred based on the learned box priors and the relation among categories in the corpus. This is a potential direction toward open-world vocabulary situations. The second way is to scale up the dataset by automatically labeling using existing foundation models such as SAM(Segment Anything Model)[1]. We will include these potential discussions to the paper.
> 2. Data scalability.
>     We are presently engaged in the expansion of the dataset by exploring SAM (Segment Anything Model)[1]. Leveraging the existing foundation models, we anticipate the creation of a more extensive and all-encompassing dataset to enhance the training of VisorGPT. Besides, leveraging text corpus in training may help to learn and infer more categories.
>
> 3. Limited application.
>     The primary objective of VisorGPT is to facilitate the generation of reasonable boxes, keypoints, and masks, serving as spatial conditions for guiding image synthesis. Additionally, there exist several sub-applications. For instance, in addition to the functions discussed in the paper, VisorGPT currently can interactively complete or refine coarse layouts provided by users, enhancing their reasonability—a crucial aspect for controllable image synthesis. This enables a more user-friendly image creation process. Furthermore, VisorGPT can be used to design printed media and application user interfaces (numerical results are shown in Table 1). We intend to incorporate these results in the latest version of the paper.
> 4. Why LLM rather than GAN or VAE? Show the advantages.
>     VisorGPT offers three key advantages:
>     i) Leveraging the transformer architecture, LLM demonstrates exceptional capabilities across various vision and language tasks.
>     ii) Utilizing language modeling enables more flexible applications by prompting such as controlling the number/size/category to be generated and scene completion shown in Fig. 3.
>     iii) Additionally, we conducted comparisons with recent state-of-the-art (SOTA) methods on designing document and mobile application interface layout namely LayoutDM (CVPR'23) [2], and VQDiffusion (CVPR'22) [3]. Notably, according to the literature, the performance of LayoutDM and VQDiffusion outperforms traditional GAN and VAE-based methods. As demonstrated in Table 1, **VisorGPT surpasses the SOTA methods on all three datasets**, including COCO, Rico[4], and PubLayNet[5]. The experimental results provide clear evidence of VisorGPT's superiority.
> 5. Synthetic data for downstream tasks.
>     One of our objectives in this paper is to aid data synthesis for downstream tasks, such as object detection or pose estimation. Accordingly, we have designed the general prompt that enables users to address data imbalance issues in such downstream tasks, such as pose estimation in crowd scenes. For instance, by specifying the number of people larger than 10 in the prompt, the generated sequences can represent crowd scenes, facilitating the synthesis of images in such contexts. However, the current layout-to-image synthesis models exhibit limitations in synthesizing images that strictly adhering to the given layout. We believe that with advancements in these models, our VisorGPT can play a substantial role in automating data synthesis, thereby contributing significantly to the field.
>
> [1] Segment Anything.
> [2] Layoutdm: discrete diffusion model for controllable layout generation.
> [3] Vector Quantized Diffusion Model for Text-to-Image Synthesis.
> [4] Rico: A Mobile App Dataset for Building Data-Driven Design Applications.
> [5] PubLayNet: largest dataset ever for document layout analysis.

---

> > ### Comment · Reviewer_6eEZ · 2023-08-15
> > **Rebuttal Acknowledgment**
> >
> > I thank the reviewers for their detailed clarification, which has addressed my concerns about their work. Therefore, I keep my rating as accepted.

---

> > > ### Author Response · Authors · 2023-08-15
> > > **Reply**
> > >
> > > Thanks for your feedback to help improve our paper. We will include the suggested contents in the latest version of our paper.

---

### Official Review · Reviewer_KtmJ · 2023-07-03

**Soundness:** 3 good
**Presentation:** 2 fair
**Contribution:** 2 fair
**Rating:** 6
**Confidence:** 3

**Summary:**

The paper proposes a method to learn a high-level generative model for images based on language modeling, which allows the generation of object masks, bounding boxes, and human pose keypoints. The paper proposes a novel tokenization scheme to turn annotations into token sequences that can be fed into an off-the-shelf language model. This tokenization scheme is targeted for these tasks and shows improved performance over alternative approaches. Quantitative experiments show that the model can capture the distribution of data on the datasets tested. Qualitative results show that the learned model is helpful for conditional image generation when compared against simple baselines for generating object bounding boxes.

**Strengths:**

- The paper leverages recent progress in language modeling to cast a generative vision problem as a language prediction problem, showing promising results that may be useful for conditional image generation.
- The paper proposes a novel way of encoding high-level image generative prior, with unique tokens designed for the three tasks studied, and they show that it improves performance over alternatives.
- Ablation studies test the main design choices. The tokenization strategy for the numbers is sound, and the paper shows that it improves over alternative strategies (for example, just tokenizing the text as done in LLM's without any special discretization for numbers or other special tokens). Ablations also show that finetuning from GPT-2 does not improve performance, with or without special tokens. Although a more powerful LLM model may overcome this, this is out of the scope of the paper, and GPT-2 seems a reasonable LLM for this ablation.
- Qualitative results show that having access to a high-level generative image prior is helpful for conditional image generation.

**Weaknesses:**

- The multimodality of the model is not used in practice nor tested. Although the method can produce keypoints, masks and boxes, from the examples and quantitative analysis it does not seem that the model is tasked to produce a mix of them in either train or test time. With this, the model may be acting as three independent models, each one being active depending on the input prompt (mask/bbox/keypoint). What is the advantage of multimodal training and can the model be prompted to produce consistent masks/bboxes and keypoints at the same time? It seems possible to train such a model as COCO contains the three modalities studied, for example.
- Related work includes a "Conditional Image Synthesis" section that is only tangentially relevant to the paper (for the qualitative applications in section 4.5). It does not include a section that analyses previous work on "Generative high-level image priors", which is the paper's main focus. There's plenty of work on generative models for images similar to the ones studied, which is disregarded, such as Scene graphs [1] and human keypoint generation [2], among others.
- Lack of comparisons to previous works. Comparisons against previous works should be performed to see how the model compares against them, especially since the model does not seem to be multimodal in practice (see points above). I.e., comparing against methods such as [1,2].
- Some results highlighted are unsurprising and do not give valuable insights, considering the progress in language modeling the paper builds upon. It is expected that the model will be able to memorize the datasets and their distribution, as shown in Figures 3 and 4, so these figures seem redundant, as they show that the model can replicate the training dataset up to a certain fidelity. A more interesting figure/analysis would be whether the model can extrapolate to configurations not seen in training. For example, if the model is prompted to produce masks and tasked to complete a sequence where a "sea" region occupies the whole canvas, is the prior for "surfboard" all of the canvas? And if the sea occupies only the left side of the frame, does the surfboard occupy only that region?
- Qualitative analysis of the trained model does not show its advantages compared to using the datasets it was trained on. How do Figure 5 and 6 show that the generative model is better than just using the datasets already available (for example, by selecting a random example that corresponds to the prompt)? An advantage would be if the model can be prompted with configurations that are not on the training set (but this is not specified in the section), or if the model can be used for completion (e.g., the prompt is five object bounding boxes, the user provides 3 instances and the model generates the other two and are consistent with the user provided ones), but these or similar settings are not studied.

[1] Scene Graph Generation: A Comprehensive Survey. Guangming Zhu, Liang Zhang, Youliang Jiang, Yixuan Dang, Haoran Hou, Peiyi Shen, Mingtao Feng, Xia Zhao, Qiguang Miao, Syed Afaq Ali Shah, Mohammed Bennamoun

[2] DGPose: Deep Generative Models for Human Body Analysis. Rodrigo de Bem, Arnab Ghosh, Thalaiyasingam Ajanthan, Ondrej Miksik, Adnane Boukhayma, N. Siddharth & Philip Torr


**Questions:**

See weakness.

**Limitations:**

Yes

---

> ### Author Rebuttal · Authors · 2023-08-10
>
> **The Figures and Tables are provided in the global pdf.**
>
> 1. The multimodal training and inference.
>
>       Thanks for pointing out. We train VisorGPT using a mix of boxes and keypoints and present some examples in Fig. 1. One can observe that VisorGPT exhibits the capability to infer bounding boxes and keypoints simultaneously. For instance in Fig. 1(a), it can deduce the presence of two individuals and a frisbee along with their corresponding bounding boxes. Besides, the keypoints for these two identified people are also predicted at the same time. The observation confirms that VisorGPT is capable of engaging in multimodal inference. The integration of multimodal training confers an advantage to the model, enabling it to acquire a cohesive representation of boxes/masks and keypoints for various objects. We will include such experiments (a mix training of bbox/mask and keypoint) and release the corresponding models.
>
> 2. Related works.
>
>       Thank you for your suggestion. We acknowledge the significance of the related works [1,2] you mentioned and have decided to incorporate them as a dedicated subsection in our paper. Furthermore, we will provide comprehensive discussions on these related works to ensure a more thorough and contextual understanding of their relevance to our study.
>
> 3. Comparison to previous works.
>
>       Thank you for your suggestion. Scene graph generation takes an image as an input and generates a scene graph, however, VisorGPT receives only the text prompts to generate boxes/keypoints. Hence. they are different and hard to compare. Instead, to further enhance the evaluation, we reproduce recent SOTA methods for designing document and mobile application interfaces such as LayoutDM(CVPR’23)[3], and VQDiffusion(CVPR’22)[4] on COCO dataset for comparison with VisorGPT. Besides, we also apply VisorGPT to documents and mobile applications design benchmarks like Rico [5] and PubLayNet [6]. As shown in Table 1, the experimental results include FID and Align. score (referrenced from LayoutDM) with lower values indicates better performance. Notably, **VisorGPT significantly surpasses these SOTA methods with better FID and Align. score on all three datasets**, showcasing the superior capability of VisorGPT to learn layouts. We will include these results and more comparisons (including pose) in the paper.
>
> 4. Extrapolation ability and concern about memorization of the dataset.
>
>       (i) To showcase VisorGPT's **extrapolation** ability, we present examples in Fig. 2. As demonstrated, when the ‘sea’ occupies the majority of the canvas, the ‘surfboard’ can appear throughout the entire canvas (Fig. 2(a)). Upon shifting the ‘sea’ region to the middle/left part of the canvas, the ‘surfboard’ is accordingly confined within the middle/left region (Figs. 2(b) and (c)). This phenomenon demonstrates that VisorGPT does not solely rely on memorization; rather, it learns intrinsic relation, location, and shape priors among categories. Consequently, **VisorGPT can extrapolate unseen/novel layouts based on users' configurations**.
>
>       (ii) To address concerns about **memorization**, we present examples in Fig.4. Firstly, we specify target classes (i.e., 'bottle', 'dining table', 'person', 'knife', 'bowl', 'oven', 'person', 'cup', 'cup', 'bowl', 'bowl', 'broccoli', 'spoon') and then select layouts satisfying these conditions from the training set (Fig. 4a, only one satisfied sample over 110K samples). In contrast, we use the same target classes as prompts for VisorGPT, inferring their bounding boxes. As depicted on the right in Fig. 4(a), the bounding boxes generated by VisorGPT exhibit diversity and considerable differences from that selected from the training set. This also validates that VisorGPT does not rely on memorizing the dataset; rather, it learns intrinsic priors among categories, facilitating the generation of novel and unseen layouts. We will include these results in the latest version of the paper.
>
> 5. Advantages compared to direct selection of bounding boxes from training set and **scene completion**.
>
>       i) When users need to specify a large number of categories (more than 10), only a few layouts in the dataset meet these requirements. For instance, in Figs. 4(a) and (b), there is only one layout in the training set containing all the required target classes. Moreover, the selected layouts from the dataset remain fixed. In contrast, as depicted on the right in Figs. 4(a) and (b), VisorGPT can receive the target classes and generate a lot of layouts with diversity that fulfill the requirements.
>
>       ii) Furthermore, our **VisorGPT facilitates interactive fulfillment of user demands**. For instance, users can provide simple layouts, and VisorGPT can subsequently augment them with additional layouts. Fig. 3 illustrates such examples. As shown in Fig. 3(a), the objective is to infer six instances ('person, person, skis, skis, skis, skis'), and users can initially provide two instances ('person, person'). VisorGPT then generates the remaining four instances (‘skis, skis, skis, skis’), with consistency to the user-provided ones. Notably, random selection from GT boxes fails to achieve such outcomes, as it cannot generate novel layouts with the diverse configurations demanded by users. Additional experimental results are presented in Appendix 1.4. We will include more examples and discussions on this matter.
>
> [1] Scene Graph Generation: A Comprehensive Survey.
> [2] DGPose: Deep Generative Models for Human Body Analysis.
> [3] Layoutdm: discrete diffusion model for controllable layout generation.
> [4] Vector Quantized Diffusion Model for Text-to-Image Synthesis
> [5] Rico: A Mobile App Dataset for Building Data-Driven Design Applications.
> [6] PubLayNet: largest dataset ever for document layout analysis.

---

> > ### Comment · Reviewer_KtmJ · 2023-08-10
> > **Rebuttal partially adresses my concerns**
> >
> > Thanks for Figure 1 and 2. I find these figures really appealing (more appealing than any other Figure in the main paper), as they show capabilities of the model that one cannot achieve by directly using the training data. I would suggest including them in the main text of the final manuscript, possibly showing distributions (e.g. with a fixed sea bounding box, what is the distribution of boards, or with a fixed snowboard, what's the distribution for person), to demonstrate they are not cherry picked examples and showing does the model break.
> >
> > The rebuttal has partly addressed some of my concerns, so I have raised the score to a weak accept, conditional to authors including the changes promised in the rebuttal. I still believe that the paper would be stronger with a more thorough analysis of its properties and showing more interesting cases of the trained model (as in Figure 1 and 2). Similarly to other authors, I do not find the results and application on image generation to be of particular interest for the proposed method. Because of these, I'm not raising the score higher.

---

> > > ### Author Response · Authors · 2023-08-11
> > > **Reply to the comments by reviewer KtmJ**
> > >
> > > Thank you for your prompt reply and raising the score, as well as your valuable suggestions. **We will re-organize the paper by presenting more of these figures including visualization of the suggested distributions and more deep analysis of these properties.** In addition, **we will explore more properties about the learned priors and add more functions of VisorGPT and give the corresponding analyses in the paper.** For instance, enabling VisorGPT to extrapolate novel layouts or modify layouts based on the given ones through users’ requests in natural language (as suggested by another reviewer), and this has been proven to be feasible in our latest experiments. Further, we will extend it to 3D space such as designing the indoor scene layouts to demonstrate the potential of VisorGPT being applied to many domains.

---

### Author Rebuttal · Authors · 2023-08-10

**The Figures and Tables are provided in the global pdf.**

---

### Author Response · Authors · 2023-08-20
**Thank ACs and all Reviewers and a Rebuttal Phase Recap**

**Thanks very much to ACs and all reviewers** for taking your time and efforts to review our paper and engage constructively with our rebuttal. As the discussion period is about to end, we would like to give a brief summary of the discussion status and outcomes.

Reviewers KtmJ, 6eEZ, xtC6, and CQCW actively participated in the rebuttal discussion phase, expressing that **their initial concerns have been addressed**. Following this positive interaction, reviewer KtmJ **maintains a rating of 7**, while reviewers 6eEZ, xtC6, and CQCW **have raised their ratings to 6**.

**We sincerely appreciate your efforts and feedback for improving our work**. To further enhance the quality of our paper, **we are committed to incorporating the suggested contents and experimental results in our final paper**.

---

### Decision · Program_Chairs · 2023-09-21

**Decision:**

Accept (poster)

**Comment:**

The paper is interesting, and its focus is timely and important, given the continuing rapid rise of visual synthesis (and their dependence of prior inputs).  Reviews raised some questions and most of them has been resolved during the rebuttal. Finally, all reviewers recommend acceptance, to varying degree. The paper will be a valuable contribution to the program.